# Scaling Laws for Optimal Data Mixtures

**Mustafa Shukor**
Sorbonne University

**Louis Bethune**
Apple

**Dan Busbridge**
Apple

**David Grangier**
Apple

**Enrico Fini**
Apple

**Alaaeldin El-Nouby**
Apple

**Pierre Ablin**
Apple

## Abstract

Large foundation models are typically trained on data from multiple domains, with the data mixture–the proportion of each domain used–playing a critical role in model performance. The standard approach to selecting this mixture relies on trial and error, which becomes impractical for large-scale pretraining. We propose a systematic method to determine the optimal data mixture for any target domain using scaling laws. Our approach accurately predicts the loss of a model of size $N$ trained with $D$ tokens and a specific domain weight vector $h$. We validate the universality of these scaling laws by demonstrating their predictive power in three distinct and large-scale settings: large language model (LLM), native multimodal model (NMM), and large vision models (LVM) pretraining. We further show that these scaling laws can extrapolate to new data mixtures and across scales: their parameters can be accurately estimated using a few small-scale training runs, and used to estimate the performance at larger scales and unseen domain weights. The scaling laws allow to derive the optimal domain weights for any target domain under a given training budget $(N,D)$, providing a principled alternative to costly trial-and-error methods.

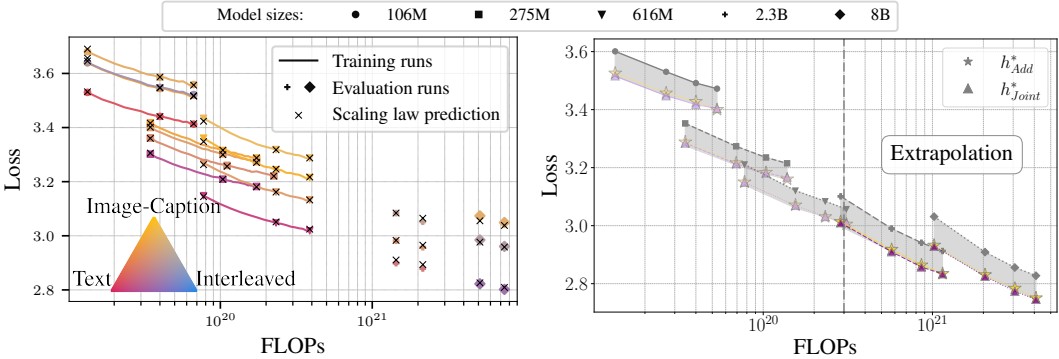

Figure 1: **Scaling Laws for Optimal Data Mixtures.** Left: We derive scaling laws that predict the loss of a model as a function of model size N, number of training tokens D, and the domain weights used to train the model (represented by the color of each point). The scaling law is fitted with small-scale runs with different domain weights, and used to predict accurately the loss of large-scale models trained with new, *unseen* domain weights. Right: We find the data mixture scaling law based on small-scale experiments (e.g., below 1B parameters) and use it to predict the optimal data mixture at larger scales (e.g., 8B parameters). Both our additive (eq. (4)) and joint (eq. (5)) laws lead to similar performance, and better than other mixtures (in the gray area). FLOPs are computed as 6ND.

39th Conference on Neural Information Processing Systems (NeurIPS 2025).

# 1 Introduction

Modern machine learning models [8, 16, 21] are pre-trained on diverse data domains, such as text for LLMs, images for vision models, and mixed modalities for multimodal models. For LLMs, these domains encompass general knowledge, code, reasoning, multilingual content, and more [5, 16, 26, 59, 60]. Multimodal models [1, 35, 44, 56, 70] are trained on a mix of text, paired, and interleaved multimodal data, and finally, large vision models are trained on image domains of different qualities, containing or not images paired with text [17, 21, 48].

The *domain weights* determine the proportion of each domain used during training, significantly impacting model performance. However, these weights are typically chosen through ad-hoc trial and error, involving training with different domain weights and selecting what works best [16, 44, 56]. Despite their critical role, a principled method for selecting domain weights is largely absent.

Scaling laws provide a theoretical framework to predict model performance. Initially developed for LLMs [29, 31, 33], these laws model the loss of a model as a function of the number of parameters $N$ and training tokens $D$. This framework has been extended to other domains and modalities [3, 56] and to account for factors such as the number of experts in mixture-of-experts models [34], sparsity [2], data repetitions [47], fine-tuning tokens [7, 69], and learning rate schedules [42].

In this work, we extend scaling laws to model the effect of domain weights on model performance. We show that the model loss depends in a predictable way on the domain weights, interacting with the number of training tokens and model parameters. We extensively validate our scaling laws in three large-scale settings: large language models (LLMs), native multimodal models (NMMs), and large vision models (LVMs) pretraining. We train large models - up to 7B parameters and 150B tokens for LLMs, 8B parameters and 160B tokens for NMMs, and 1B parameters with 330B tokens for LVMs. across multiple text, multimodal, and image domains. The key takeaways from our work are:

**Scaling laws that extrapolate.** We demonstrate that our scaling laws can be fitted using small-scale runs, and then provide an accurate prediction of the loss of large-scale models trained with new, *unseen* domain weights. This is illustrated by Fig. 1, left, where we accurately predict the loss of models trained with an order of magnitude more compute than the small-scale runs.

**Optimal domain weights estimation.** Once fitted, these scaling laws give an accurate estimation of the loss as a function of the domain weights, which is minimized to give optimal domain weights. This provides a principled alternative to the costly practice of trying different domain weights and selecting the best one. This is illustrated by Fig. 1, right, where we report the average loss of NMMs.

This paper is organized as follows. In Sec. 2, we introduce the problem of domain weight selection and describe our scaling law formulations. In Sec. 3, we detail the model architectures and data domains for LLM, NMM, and LVM pretraining. Sec. 4 demonstrates that our scaling laws accurately extrapolate to new domain weights, larger model sizes and number of tokens. Sec. 5 shows how the fitted laws can be used to estimate the optimal domain weights from a few small-scale runs. Finally, Sec. 6 explores various aspects of our scaling laws, including quantifying the number of runs needed to get a satisfying estimation, how the best domain weights change when scaling flops, as well as alternative scaling laws formulations. Finally, we discuss related works in Sec. 7.

# 2 Data mixture scaling laws

## 2.1 Problem setup

We consider training models with data coming from $k$ data domains $\mathcal{D}_1, \ldots, \mathcal{D}_k$; we can query random samples $x$ from any domain $\mathcal{D}_i$. Consequently, we can sample from the mixture $\mathrm{mix}(h) = \sum_{i=1}^{k} h_i \mathcal{D}_i$ for any *domain weights* $h$, following the law $p(x|\mathrm{mix}(h)) = \sum_{i=1}^{k} h_i p(x|\mathcal{D}_i)$. Here, $h$ is a $k-$dimensional vector of positive entries that sum to one, that is, an element of the *simplex* $\Delta_k$. In plain words, data is sampled from the domain $i$ with probability $h_i$. We have a target domain $\mathcal{D}_T$, which may — but need not — be one of the training domains. We consider a model with $N$ parameters, represented with the vector of parameters $\theta \in \mathbb{R}^N$. Finally, we have a loss function $\ell(x, \theta)$ defined for any $x$ in the data domains $\mathcal{D}_i$ or the target domain $\mathcal{D}_T$. This defines the loss for any domain weights $h$, as well as the target loss, as the expectations

$$L_h(\theta) = \mathbb{E}_{x \sim \mathrm{mix}(h)} [\ell(x, \theta)] \text{ and } L_T(\theta) = \mathbb{E}_{x \sim \mathcal{D}_T} [\ell(x, \theta)] \ . \tag{1}$$

The training of the model, with fixed domain weights $h$, is done by running an optimization algorithm such as Adam to approximately minimize $L_h$. In the course of its execution, the optimization algorithm processes $D$ tokens and outputs trained parameters $\theta^*(h, D)$ of size $N$. The goal of this paper is to predict the loss on the target domain $\mathcal{D}_T$ after training a model of size $N$ with $D$ tokens with domain weights $h$; a quantity denoted as $\mathcal{L}(N, D, h)$ defined as $L_T(\theta^*(h, D))$. In practice, we can, of course, have several target domains that capture different aspects of a model's capabilities. In that case, we estimate the target loss on all of the target domains by fitting multiple scaling laws.

This framework is sufficiently general to encompass various model architectures and modalities. In this work, we consider different domains to be either various text domain datasets, various image domains, different modalities (e.g., image and text), or different data types (e.g., paired or interleaved).

## 2.2 Scaling laws derivation

In their original form, scaling laws predict the *training loss* of a model of size $N$ trained with $D$ tokens [33]. The Chinchilla scaling law models training loss as an *additive power law* [31]:

$$\mathcal{L}(N, D) = E + \frac{A}{N^\alpha} + \frac{B}{D^\beta} \ , \tag{2}$$

where $E, A, \alpha$ and $\beta$ are parameters that depend on the training set, model's architecture, and optimization algorithm. We depart from these original scaling laws in two ways: i) we consider the loss of a model on a *target domain* that need not be the training domain, and more importantly, ii) we quantify the impact of the domain weights $h$ on the loss. Regarding i), as already been shown in several works [24, 29, 46, 56], the loss on a target domain can still be modeled by a scaling law of the form eq. (2). Hence, for every domain weights $h$ used for training, we expect the loss on the target domain to follow a Chinchilla power law, where the coefficients depend on $h$. In other words, the loss on the target domain can be expressed as:

$$\mathcal{L}(N, D, h) = E^h + \frac{A^h}{N^{\alpha^h}} + \frac{B^h}{D^{\beta^h}}. \tag{3}$$

The question now is, how do the parameters $E^h, A^h, \alpha^h, B^h$ and $\beta^h$ depend on $h$? We propose two different formulas that use simple formulas for these parameters. We first study the *additive* scaling law, in which $E^h$ is modeled as a function of $h$, while the other parameters are taken as constants:

$$\mathcal{L} = E + \frac{1}{\sum_{i=1}^k C_i h_i^{\gamma_i}} + \frac{A}{N^\alpha} + \frac{B}{D^\beta}. \tag{4}$$

The parameters of the scaling law are $Z = (E, A, B, \alpha, \beta, (C_i)_{i=1}^k, (\gamma_i)_{i=1}^k)$, which depend on the model architecture, the target and the source domains. This scaling law has $5 + 2k$ parameters. Since this scaling law is additive, the optimal domain weights $h^*$ that minimize it are independent of the model size $N$ and the number of tokens $D$. In order to capture the interaction between scale and mixture, we also propose the *joint* scaling law:

$$\mathcal{L} = E + \frac{1}{\sum_{i=1}^k C_i h_i^{\gamma_i}} + \frac{A^h}{N^\alpha} + \frac{B^h}{D^\beta} \text{ with } A^h = (\sum_{i=1}^k C_i^A h_i)^{\gamma^A} \text{ and } B^h = (\sum_{i=1}^k C_i^B h_i)^{\gamma^B} \tag{5}$$

In that scaling law, we consider the same dependency in $h$ for the bias term $E$ as in the eq. (4), and we additionally model the terms $A^h$ and $B^h$ as simple functions of $h$. The parameters of the law are $Z = (E, \alpha, \beta, (C_i)_{i=1}^k, (\gamma_i)_{i=1}^k, (C_i^A)_{i=1}^k, \gamma^A, (C_i^B)_{i=1}^k, \gamma^B)$, which gives $5 + 4k$ parameters. In this law, there is an interaction between $N$, $D$ and $h$, in the sense that $\frac{\partial^2 \mathcal{L}}{\partial N \partial h} \neq 0$ and $\frac{\partial^2 \mathcal{L}}{\partial D \partial h} \neq 0$, while these partial derivatives are $0$ for the additive scaling law. This law predicts that the contribution of N and D to the loss depends on the domain weights, and as such, the optimal domain weights are compute-dependent. The joint scaling law is more expressive than the additive scaling law, since we can recover eq. (4) by taking $\gamma^A = \gamma^B = 1$ and $C_i^A = A, C_i^B = B$ for all domains $i$. As such, if the scaling laws are fitted properly, the error on the training runs is always lower for the joint scaling law than for the additive scaling law. The joint scaling law still models the terms $\alpha^h$ and $\beta^h$ as constants. We tried modeling these terms using the same simple parametric form depending on $h$, but it only yields minor improvements (see Sec. 6). On the other hand, going from the additive to the joint law often decreased the estimation error. Hence, we restrict the bulk of our study to these two laws.

## 2.3  Fitting the scaling laws

In order to fit the scaling laws, we launch several training runs with different domain weights $h$, model sizes $N$, and number of tokens $D$, and record the loss on the target domain $L_T$. We train model sizes and number of tokens that are evenly spaced. We chose the training domain weights by taking a grid of evenly spaced points in the simplex, where each domain weight is above a minimal value (*i.e.*, 0.1). We have $p$ input-target pairs $(N^j, D^j, h^j), L_T^j$ for $j = 1, \ldots, p$ where $p$ is the number of training runs. The optimal parameters $Z^*$ are obtained by minimizing the standard Huber loss:

$$H(Z) = \frac{1}{p} \sum_{j=1}^{p} \text{Huber} \left( L_T^j - \mathcal{L}(N^j, D^j, h^j; Z) \right), \quad (6)$$

where $\text{Huber}(x) = \frac{x^2}{2}$ if $|x| < \delta$ and $\delta(|x| - \frac{\delta}{2})$ otherwise, where $\delta$ is a hyperparameter which we take as $\delta = 1\text{e-3}$.

The standard technique to fit scaling laws consists of using L-BFGS [39] to minimize the loss starting from an evenly-spaced grid of initial parameters $Z$ and then retaining the smallest local minimum. Contrary to most scaling laws that involve only 5 parameters, our scaling laws have $5 + 2k$ or $5 + 4k$ parameters, where $k$ is the number of domains. In our experiments, we use up to $k = 8$ domains, which gives 37 parameters to fit. This increased dimensionality makes the standard technique to fit scaling laws cumbersome. We propose two changes that lead to good fit. Firstly, we use a random search, to sample the initial parameters $Z$. Secondly, we use the Basin-hopping algorithm [63] instead of L-BFGS to explore the minimizers of the loss function. The Basin-hopping algorithm itself uses L-BFGS as an inner routine to minimize the loss function,

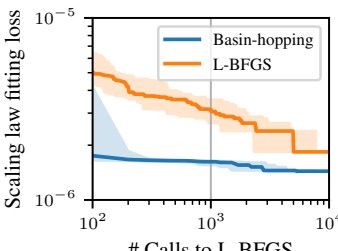

Figure 2: Value of the Huber loss (6) as a function of the number of L-BFGS calls to fit eq. (5) on the Interleaved domain from the multimodal experiment ($p = 1062$ input-target pairs, $k = 3$ domains). We repeat 100 random trials, the bold line is the median, and the shaded regions are the 25-75% quantiles. The Basin-hopping method with L-BFGS subroutine converges faster than repeated calls to L-BFGS.

but it also uses a random walk to explore the space of local minima. Fig. 2 gives an example of the performance of the algorithm: to reach a low fitting loss, the Basin-hopping algorithm requires far fewer calls to L-BFGS than doing a random search over the L-BFGS initializations.

In order to evaluate the scaling law, we take a new set of runs that give different values of $(N, D, h)$, and compare the loss on those runs predicted by the scaling law against the actual loss achieved by the model. We quantify this with the Mean Relative Error (MRE), computed as |prediction − observation|/observation, and we report it as a percentage.

## 3  Experimental setup

We give an overview of the models and domains used in our experiments. Detailed architectures and hyperparameters are given in appendix A.

### 3.1  Pretraining of large language models (LLMs)

**Models.** We use transformers [62] for autoregressive language modelling. We use the same setup as llama [61], with rotary positional embeddings, SwiGLU activations, and RMSNorm. The models are scaled by changing the latent dimension, with model sizes ranging from 186M to 7B parameters.

For some smaller-scale analyses, we also use GPT2-style transformers [51] to perform autoregressive language modeling with model sizes ranging from 90M to 3B parameters.

**Datasets.** For the main experiments, we use the $k = 7$ domains from slimpajama [57]. We use these domains as distributed by the authors, without any additional data filtering. For some smaller-scale analyses, we use up to $k = 8$ domains coming from the Pile dataset [22]: Wikipedia, StackExchange, GitHub, pg19, arxiv, free law, openwebtext, and PubMed Central.

## 3.2 Pretraining of native multimodal models (NMMs)

**Models.** We pretrain native multimodal models (NMMs), based on an early-fusion architecture [6, 56] and follow the design and implementation proposed in [56]. The model consists of a single transformer [62] without a separate vision encoder, resulting in the same architecture used for LLMs. The model processes a sequence of interleaved text and image tokens. Text tokens are obtained using a standard LLM tokenizer, while image tokens are obtained by patchifying the image and applying a linear projection. Images are resized to 224×224 resolution with a 14×14 patch size. The overall model architecture is aligned with [36], incorporating SwiGLU FFNs [54] and QK-Norm [15].

**Datasets.** Following previous works [35, 38, 56] we train on a mixture of multimodal datasets, covering $k = 3$ data types: (1) text-only data from DCLM [36], (2) interleaved multimodal documents from Obelics [35], and (3) paired image-caption datasets from DFN [20], COYO [10], and a private collection of High-Quality Image-Text Pairs (HQITP).

## 3.3 Pretraining of large vision models (LVMs)

**Models.** We pretrain large vision models with a multimodal objective, following the AIMv2 recipe [21]. Unlike traditional language modeling or multimodal models described above that focus on text decoding, AIMv2 trains a vision encoder using an autoregressive objective on both image and text tokens. The model architecture is composed of a vision encoder and a multimodal decoder stitched together in a late-fusion fashion.

**Datasets.** We train on a mixture of paired of image-caption drawn from four domains ($k = 4$): (1) noisy alt-text including COYO-700M [10] and DFN2B [20], which provide large-scale real-world image-text pairs with varying levels of noise and quality; (2) HQ-ITP-1, a high-quality dataset containing 134 million samples; (3) HQ-ITP-2, another high-quality dataset comprising 400 million samples; and (4) synthetic data, consisting of recaptioned versions of DFN2B and HQ-ITP-2.

## 3.4 Implementation details

In order to scale models, we change the hidden dimension size in the transformers $d$, keeping a fixed number of layers. To reduce the experimental cost, most of the experiments are done with a constant learning rate scheduler. This allows us to collect many points with varying numbers of tokens $D$ for each run, rather than just one per run. We also validate our findings when using cosine learning scheduler in appendix C.1, where we show that the scaling laws also extrapolate from small-scale to large-scale behavior in that case.

## 4 Predicting large-scale performance from small-scale experiments

In this section, we demonstrate that (a) our scaling laws accurately capture the training data, and (b) generalize effectively to larger scales with significantly increased values of N and D. To this end, we fit the laws using small models trained with a small number of tokens, and we validate them on larger models with a large number of tokens. We experiment with LLMs, trained with a mixture of text domains, NMMs, trained with a mixture of multimodal domains, and LVMs, trained with images of different qualities and paired or not with text. For LLMs, we consider the $k = 7$ domains from slimpajama, which are different text domains. For multimodal pretraining, and similar to previous works, [35, 56, 70], the data mixture spans $k = 3$ different domains: text, paired (image-captions), and interleaved multimodal data. For large vision model pretraining, we use $k = 4$ domains. Tab. 1 displays the different model sizes, number of training tokens, and number of different domains weights that we use to train and evaluate the scaling laws.

**Results.** Fig. 3 presents a comparison between the actual loss achieved by our trained models and the loss predicted by our scaling laws. We summarize the results on each domain by showing the average predicted loss (full results in appendix B). Remarkably, the predicted losses align closely with the observed values for both the joint and additive laws. In addition, the laws show good extrapolation to larger model sizes. To further quantify this alignment, we report the mean relative error (MRE%) in Tab. 2, which reveals a consistently low MRE% for both laws, with an improvement of the joint law over the additive one. These results demonstrate that we can fit the scaling laws on small scales and extrapolate to larger scales. We remark that the MREs have some degree of variability across

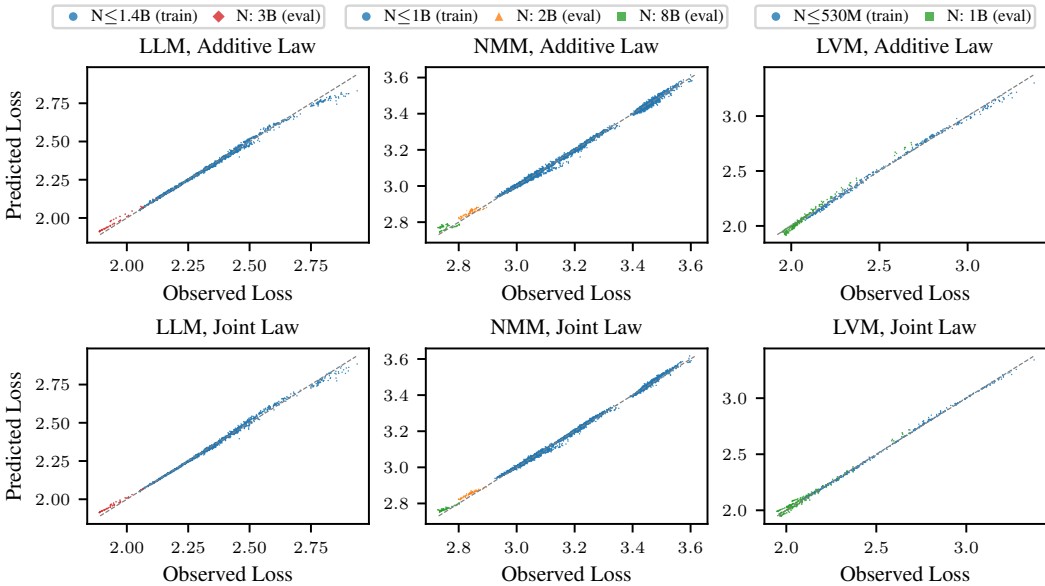

Figure 3: **Observed vs predicted loss** for LLM pretraining on domains from the slimpajama dataset, NMM pretraining with multimodal domains, and LVM pretraining with image-caption domains. The scaling laws are fitted on small-scale models (blue points in the figure) and extrapolated to larger models. We display here the average loss over all domains for each modality. The MRE% for each domain is reported in Tab. 2.

Table 1: **Experimental setup** for the extrapolation experiment.

|  |  | N | D | # domain weights | FLOPs |
|---|---|---|---|---|---|
| LLM | Train | 412M | 4-20B | 60 | $2.5 \times 10^{22}$ |
|  |  | 834M | 7-26B | 40 |  |
|  |  | 1.1B | 7-36B | 40 |  |
|  |  | 1.4B | 13-46B | 20 |  |
|  | Eval | 3B | 20-100B | 4 | $7.2 \times 10^{21}$ |
|  |  | 7B | 150B | 1 | $6.3 \times 10^{21}$ |
| NMM | Train | 106M | 20-100B | 32 | $3.7 \times 10^{22}$ |
|  |  | 275M | 20-100B | 31 |  |
|  |  | 616M | 20-100B | 30 |  |
|  |  | 932M | 20-100B | 34 |  |
|  | Eval | 2.3B | 100-160B | 8 | $1.8 \times 10^{22}$ |
|  |  | 8B | 100-160B | 4 | $3.1 \times 10^{22}$ |
| LVM | Train | 89M | 17-50B | 32 | $1.4 \times 10^{22}$ |
|  |  | 157M | 17-50B | 32 |  |
|  |  | 306M | 17-100B | 15 |  |
|  |  | 531M | 17-170B | 16 |  |
|  | Eval | 1.1B | 17-334B | 8 | $1.8 \times 10^{22}$ |

Table 2: **Scaling laws MRE**

| Modality | Target domain | MRE% (Additive / Joint law) | |
|---|---|---|---|
|  |  | Train | Val |
| Language | Arxiv | 0.50 / 0.39 | 2.09 / **1.62** |
|  | book | 0.29 / 0.24 | **0.80** / 1.19 |
|  | C4 | 0.29 / 0.24 | **0.31** / 0.34 |
|  | Github | 0.65 / 0.54 | **1.17** / 2.51 |
|  | Commoncrawl | 0.29 / 0.24 | **0.58** / 0.90 |
|  | Stackexchange | 0.51 / 0.38 | **0.36** / 0.47 |
|  | Wikipedia | 0.92 / 0.57 | 4.45 / **2.09** |
| Multimodal | Image-Caption | 0.47 / 0.43 | **0.95 / 0.95** |
|  | Interleaved | 0.14 / 0.10 | 0.48 / **0.42** |
|  | Text | 0.12 / 0.10 | 0.44 / **0.37** |
| Vision | Noisy image-text | 0.35 / 0.23 | 1.20 / **0.63** |
|  | Synthetic | 1.89 / 0.83 | 6.19 / **5.94** |
|  | High-quality 1 | 1.19 / 0.34 | 2.02 / **1.18** |
|  | High-quality 2 | 0.64 / 0.31 | **0.79** / 1.06 |

domains; for instance, on the LLM experiment, we get at the same time an extremely low MRE of 0.31% on the C4 dataset and a high MRE of 4.45% on Wikipedia.

**FLOPs count.** We report the approximate computational cost of running the small-scale experiments to fit the scaling laws, and compare it to the cost of large-scale runs in Tab. 1. We compute the FLOPs required for one run as $6ND$. The cost of large-scale runs is comparable to that of small-scale runs, and the large-scale runs could be trained with far more tokens, which would increase their FLOPs.

## 5 Optimal data mixtures

**Optimal domain weights estimation** Once the scaling law is fitted, we can derive the optimal domain weights $h^*$ that minimize $\mathcal{L}(N, D, h)$ under the simplex constraint $h \in \Delta_k$. We solve it using mirror descent, i.e. iterating $h^{t+1} = \frac{\hat{h}^t}{\sum_i \hat{h}^t_i}$ where $\hat{h}^t = h^t \times \exp(-\eta \nabla_h \mathcal{L}(N, D, h))$, with $\eta$

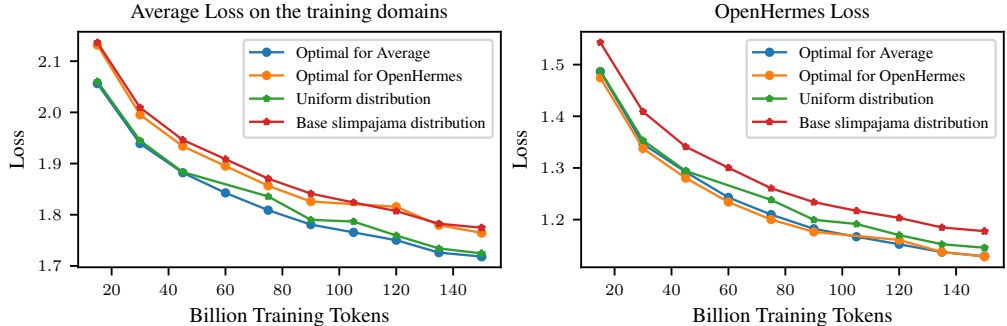

Figure 4: **Losses of the 7B models.** After fitting the scaling laws on the small scale runs, we estimate the optimal domain weights $h^*_{avg}$ that minimize the average loss over the training domains (left), and $h^*_{OH}$ that minimizes the loss on the OpenHermes dataset (right). We then train 7B models with these optimal weights, and compare them to two baselines: one with uniform weights, and one with the standard distribution of slimpajama. The losses are averaged over all training domains, and also reported on the OpenHermes dataset. As expected, the model trained with $h^*_{OH}$ performs best on OpenHermes, while the model trained with $h^*_{avg}$ performs best on the training domains.

a small step size. In practice, we may want to obtain a model that works well on several tasks at once, with weights $w$. In that case, we have $m$ different target domains $\mathcal{D}^1_T, \ldots, \mathcal{D}^m_T$. We estimate the scaling law for each target domain $\mathcal{D}^i_T$ and obtain $m$ different scaling laws $\mathcal{L}^i(N, D, h)$. The optimal domain weights $h^*$ that are good on average are given by:

$$h^*(N, D) \in \arg\min_{h \in \Delta_k} \sum_{i=1}^{m} \mathcal{L}^i(N, D, h). \tag{7}$$

The behavior of $h^*$ depends on the scaling law that we consider. Since eq. (4) assumes an additive relationship, the minimizer of eq. (7) is *independent* of $N, D$; in other words, it does not depend on scale. On the other hand, eq. (5) takes into account a multiplicative interaction between $N, D$ and $h$. Therefore, the optimal $h$ is scale-dependent. If one task is more important than another, we can incorporate importance weights in the sum in eq. (7).

The main practical takeaway of our paper is this simplified approach to optimal mixture estimation. Indeed, as demonstrated in Sec. 4, we can accurately fit our scaling laws with small-scale runs. Using these scaling laws, we can then solve eq. (7) for various targets $(N, D)$, which gives a principled way of choosing domain weights, rather than using ad-hoc methods as usually done in practice. To demonstrate our point, we do this for different modalities considered in the paper

**LLM results.** Since the additive scaling law gave us the lowest MREs, we use it to estimate the optimal data mixture that minimizes the average loss over the $k = 7$ training domains, which we denote $h^*_{avg}$ We then train a 7B model with 150B tokens with that optimal data mixture.

For all the runs, we also monitor the loss on the OpenHermes dataset, which is a small high-quality dataset used for model alignment. We fit the scaling laws for that domain as well, even though this domain is *not* part of the pre-training domains. The rationale is that we want to estimate weights that lead to the best performance on this high quality dataset, which should be a proxy of performance on downstream tasks. We then find the optimal domain weights for that scaling law, which we denote $h^*_{OH}$, and train another 7B model with 150B tokens. As baselines, we train two more 7B models with that many tokens, one with the standard distribution of slimpajama, proportional to the number of tokens in each domain, and one with a uniform distribution over domains.

Since we want the best models possible, we use a cosine learning rate schedule, which makes the scaling law extrapolation impossible to conduct. We report the average loss of these models on the training domains, on the OpenHermes dataset in Figure 4. We also evaluate them on many downstream tasks and report the results in Table 3. The model trained with weights $h^*_{OH}$ is overal better than the other models in terms of evaluations. We believe that the pipeline demonstrated in this paper — estimate the scaling law for the loss on a high quality domain with small scale runs, find the minimizer, train a large scale model with it — is a promising avenue to get better models.

Table 3: **Evaluations of the 7B models.** We report the median score on the CORE tasks [36] as well as other accuracies on standard benchmarks. The model trained with weights that attempt to minimize the loss on OpenHermes performs best.

| Weights | CORE | MMLU | Arc-easy | Arc-challenge | Boolq | Piqa | Siqa | Hellaswag | Winogrande |
|---------|------|------|----------|---------------|-------|------|------|-----------|------------|
| $h^*_{avg}$ | 56 | 32 | **71** | 40 | 60 | 79 | 52 | 72 | 65 |
| $h^*_{OH}$ | **58** | **37** | 70 | 40 | **67** | 79 | 53 | 72 | 65 |
| Uniform | 53 | 30 | 70 | 40 | 46 | 78 | 51 | 70 | 65 |
| Base | 52 | 25 | 70 | **43** | 51 | 79 | 53 | 71 | 65 |

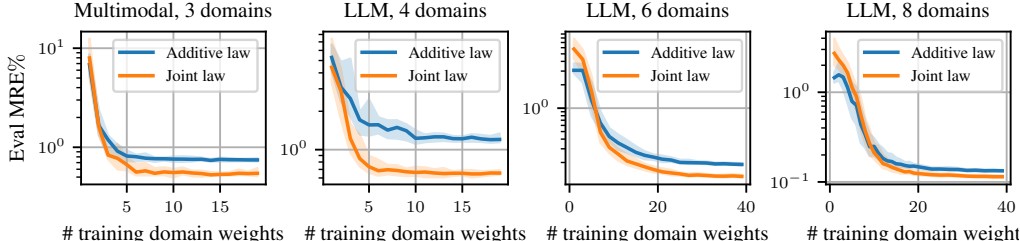

Figure 5: **Evaluation of the scaling law as a function of the number of training runs.** We randomly select $q$ different domain weights $h_{\text{train}} = [h_1, \ldots, h_q]$, and only use the runs that use these histograms to fit the scaling laws. We then evaluate the MRE on all the domain weights $h_{\text{test}}$ that are not part of $h_{\text{train}}$. For the multimodal and LLM with 4 domains, we compute the eval MRE on the large-scale (resp. 1B and 8B) models. For the LLM with 6 and 8 domains, we compute the eval MRE on same size models.

**NMM results.** We fit both the additive and joint scaling laws on three multimodal data domains, using only small models. For the joint scaling laws, we predict the best training mixture $h^*$ that minimizes the average of the domain losses for each model size while fixing the number of tokens at 100B for practicality. We then train models with these optimized mixtures. Fig. 1 compares the performance of these best mixtures against uniform mixtures, those used in prior works [44, 56], and randomly sampled mixtures that cover an important area of mixture grid. Models trained with our estimated mixtures consistently outperform alternatives. Notably, both additive and joint laws perform similarly well, making additive laws a strong and more practical baseline, since it takes the same optimal mixture for all runs. Remarkably, the optimized mixtures generalize effectively to larger model sizes, which validates the possibility of choosing the optimal mixture based on small-scale experiments, and then extrapolating to larger scales.

**LVM results.** We fit the scaling laws on the AIMv2 data mixture, which consists of 4 domains, and we estimate the optimal domain weights that minimize the average loss over these domains. We then train a 1B model with these optimal weights, and compare it to a model trained with uniform weights. We find that the model trained with the optimal weights performs better than the one trained with uniform weights, which validates our approach once again.

## 6 Scaling laws analysis

In that section, we conduct LLM experiments in a different setting, using the Pile dataset [22], which allows us to have a variable number of domains, between $k = 4$ and $k = 8$.

**Only 10-20 runs are needed to fit the scaling laws.** We investigate the number of runs needed for an accurate scaling law fitting. We randomly partition the domain weights into $h_{\text{train}} = [h_1, \ldots, h_q]$, of size $q$, and put the other domain weights into $h_{\text{test}}$. We fit the scaling law on $h_{\text{train}}$ and report the MRE on the test domain weights. Since the number of parameters of the scaling law depends linearly on the number of domains $k$, we expect the number of runs necessary to fit the scaling law to increase when we consider more domains. To verify this hypothesis, we consider the NMM pretraining experiments, with $k = 3$ domains and LLM pretraining with $k = 4, 6, 8$ domains. For the NMM with 3 domains and LLM with 4 domains as considered so far, we fit the laws on small-scale models and compute the MRE on large-scale models as in Sec. 4. For the LLM training with $k = 6, 8$ domains,

because of the very large search space with a high number of domains, we take a single model size and skip the dependency on $N$ in the scaling law, only considering the dependency on $h$ and the number of training tokens $D$. We report the MRE as a function of the number of training histograms $q$ in Fig. 5. We observe that we need about 10 runs for the NMM and LLM with 4 domains to get to an optimal MRE, while we need about 20 for LLM with 6 and 8 domains. Interestingly, we observe that when the number of training runs is very low, the additive law has a slightly lower eval MRE, due to its lower number of parameters.

**Optimal domain weights behavior when scaling FLOPs.** We study how the optimal mixture $h^*(N, D)$ for the average loss evolves as a function of the compute-budget $(N, D)$ on the multimodal models, as predicted by the joint scaling law. We report results in Fig. 6. We see that interleaved data gets less important as we increase $D$, whereas bigger models tend to rely more on text. The additive law captures the average behavior across all scales.

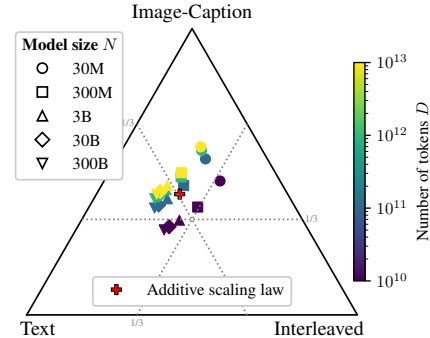

Figure 6: **Evolution of optimal domain weights** $h^*$ **with compute budget** $(N, D)$ on the multimodal data, as predicted by the joint scaling law (eq. (5)).

**Other scaling laws formulas.** We investigate alternative scaling laws, and validate our proposed scaling laws, by evaluating other formulas in the same setup as in Sec. 4. First, we want to understand whether we could use a simpler form for the dependency on the domain weights $h$. To do so, we use the "simple additive" scaling law

$$\mathcal{L} = E + (\sum_{i=1}^{k} C_i h_i)^\gamma + \frac{A}{D^\alpha} + \frac{B}{N^\beta}, \qquad (8)$$

where the dependency in $h$ is simpler compared to the additive and joint scaling laws. This law has $k-1$ fewer parameters than the additive scaling law. The joint scaling law models the terms $A^h$ and $B^h$ as functions of domain weights. We want to understand whether also taking a dependency of $\alpha$ and $\beta$ on $h$ helps capture more information about models' behavior. To do so, we consider the "full" scaling law:

$$\mathcal{L} = E + \frac{1}{\sum_{i=1}^{k} C_i h_i^{\gamma_i}} + \frac{A^h}{N^{\alpha^h}} + \frac{B^h}{D^{\beta^h}}, \text{ with} \qquad (9)$$

$$A^h = (\sum_{i=1}^{k} C_i^A h_i)^{\gamma^A}, \quad B^h = (\sum_{i=1}^{k} C_i^B h_i)^{\gamma^B}, \quad \alpha^h = (\sum_{i=1}^{k} C_i^\alpha h_i)^{\gamma^\alpha} \text{ and } \beta^h = (\sum_{i=1}^{k} C_i^\beta h_i)^{\gamma^\beta} \quad (10)$$

This law is more expressive than the joint scaling law, and it adds $2k + 1$ parameters. We give the full results of those laws in appendix B, and report the average MRE in Tab. 4. We see that the additive law reduces the MRE a lot compared to the simple law, especially in the LLM experiment. Despite having a slightly smaller train MRE, the full law does not extrapolate as well as the joint law. For the LVM experiment, the picture is different: the simple law works better than the additive law, and the full scaling law brings some improvement over the joint law.

Table 4: **Other Scaling laws average MRE %**

|  | Simple | **Additive** | **Joint** | Full |
|---|---|---|---|---|
| NMM | 0.70 | 0.62 | 0.58 | 0.60 |
| LLM | 1.70 | 1.39 | 1.30 | 1.21 |
| LVM | 2.31 | 2.55 | 2.21 | 2.04 |

## 7   Related works

**Scaling laws.** Scaling laws research investigates how model performance varies with training compute. Foundational studies [30, 31, 33] established that language models follow a predictable power-law relationship between performance and compute, allowing for the optimal allocation of parameters and training tokens within a specified budget. Scaling behavior has since been explored

across a wide range of domains, including vision models [21, 53], diffusion transformers [37], and other fields [12, 49]. While typical scaling laws consider the number of total parameters, other studies have examined the influence of both width and depth [45], or the number of parameters allocated to the teacher and student in cse of model distillation [9]. Sparse Mixture of Experts (MoE) models have been another focus, with investigations into how factors like sparsity, the number of experts, and routing strategies affect scaling [2, 14, 34, 64]. For multimodal models, scaling laws have been explored in studies such as [3, 56]. Of particular relevance is [56], which examines native multimodal models. However, their analysis is constrained by a fixed pretraining mixture.

**Scaling laws for data mixtures** Optimizing data mixtures for model training is a critical challenge, often requiring extensive experimentation. Recent studies have begun exploring systematic approaches to identify optimal mixtures more efficiently. For instance, Goyal et al. [25] investigated scaling laws for data filtering in CLIP models, emphasizing data quality and repetition. Gu et al. [28] examined scaling laws for continual pretraining of language models, predicting the optimal balance between pretraining and domain-specific data, while Bethune et al. [7] followed the same approach, focused on forgetting in finetuning. Similarly, [11] derived scaling laws that account for data quality factors such as diversity. Closer to our work, Ye et al. [68] and Ge et al. [23] propose scaling laws that model the loss as a function of $h$ for fixed (N, D), but they do not consider a joint law for $(N, D, h)$ as we do here. We also find that, in our experiments, for a fixed (N, D), our scaling laws extrapolate better to unseen mixtures (see appendix B). Further, these approaches are generally constrained to a single modality, and they consider relatively small models, below 1B parameters.

**Data mixture selection.** The standard approach to selecting training data mixtures relies on trial and error, where different combinations are tested to determine the best-performing mixture [21, 35, 38, 44, 55, 58, 70]. However, this method is costly, leading to recent efforts exploring alternative strategies. Some studies adopt heuristic methods, adjusting mixture ratios based on data sizes for each domain [13, 27, 52] or to match a target task's distribution [26]. Others predict model performance using small models that take the mixture as input [4, 19, 40, 67]. A third approach employs auxiliary models to rank and select high-quality training data, which has been popularized recently by large foundation models [8, 16, 50, 65, 66].

## 8 Discussion

**Limitations.** Our current study is focused on pretraining, but continual pre-training and finetuning are also scenarios in which the mixture is important. Our scaling law predicts a generic target loss [33], which is known to correlate with downstream task performance [31, 43]. Future work may involve predicting this performance directly, like [32]. Furthermore, assuming no data repetition (i.e., an infinite stream of data from each domain), as is typical for LLM pretraining, is unrealistic when training with very scarce, high-quality sources. Finally, we assume that the mixture is fixed throughout training, but future works may consider a dynamic evolution of the weights (*e.g.*, curriculum learning).

**Broader impact.** Mixture coefficients have a tremenduous impact on the performance on downstream tasks. Modern training corpora are typically a combination of dozens of sub-domains, striking a balance between diversity and quality. Giving the cost of pre-training, finding the optimal mixture through extensive trials and errors can be prohibitively expensive. Our scaling law only require a few runs at small scale to yield meaningful coefficients for larger models. Our work also has environmental benefits, as it significantly reduces the cost of pre-training, including the amount of $CO_2$ emission and the energy needed. Moreover, it may yield better models in the long run.

**Conclusion.** We propose scaling laws that predict the loss on an arbitrary target domain, from both mixture coefficients and the compute budget $(N, D)$. Our laws hold for language, multimodal, and vision pretraining. The optimal mixture coefficients found from a small scale can be used for much larger models and training budgets, demonstrating significant improvement over domain weights found by naive grid search. This work paves the way to a principled theory of data mixture selection.

## Aknowledgment

We thank Victor Guilherme Turrisi da Costa for technical support. We thank Marco Cuturi, Jason Ramapuram, Denise Hui, Samy Bengio, and the MLR team at Apple for their support throughout the project.

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

| Params | 412M | 834M | 1.1B | 1.4B | 3B | 7B |
|---|---|---|---|---|---|---|
| width | 1024 | 1536 | 1792 | 2048 | 3072 | 4864 |
| depth | | | 24 | | | |
| Learning rate | 6e-4 | 4e-3 | 3.4e-4 | 3e-4 | 2e-4 | 1.2e-4 |
| Optimizer | | | Fully decoupled AdamW [41] | | | |
| Optimizer Momentum | | | $\beta_1 = 0.9, \beta_2 = 0.95$ | | | |
| Minimum Learning rate | | | 0 | | | |
| Warm up iterations | | | 2k | | | |
| Weight decay | | | 1e-4 | | | |
| Batch size | | | 2M | | | |

Table 5: **Pre-training hyperparameters** used for pre-training of LLM to conduct the main scaling laws study.

# A   Implementation details

## A.1   LLM pretraining

For the main experiments, we use LLAMA style architectures. For the analyses with the Pile domains, we borrow architectures from [8]. We use a fixed depth of 24 and change the latent dimension of the network to obtain different model scales. All hyperparameters are described in Tab. 5 and Tab. 7.

## A.2   Multimodal pretraining

**Implementation details.**   We closely follow the implementation of [56] and present in Tab. 6 the pre-training hyperparameters for the model configurations used in our scaling laws study. Training is conducted in the L3M code base [18]. Our models range from 100M to 8B parameters, with width scaling accordingly while maintaining a constant depth of 24 layers. We use causal attention for text tokens and bidirectional attention for image tokens. Learning rates are adjusted based on model size, generally decreasing for larger models. These values were determined through empirical testing. Optimization is handled using a fully decoupled AdamW optimizer with momentum values set to $\beta\_1 = 0.9$, $\beta\_2 = 0.95$, and a weight decay of $1 \times 10^{-4}$. Each training batch consists of 2,000 samples, totaling 2 million tokens with a 1,024-token context length. Gradients are clipped at 1.0, and training begins with a warmup phase of 1,000 iterations, followed by a constant learning rate schedule to reduce the number of experiments.

For vision inputs, we process images as $(14, 14)$ patches with augmentations including Random Resized Crop (224px, scale range of [0.4, 1.0]) and Random Horizontal Flip with a 50% chance. Model training benefits from efficiency techniques such as `bfloat16` precision, Fully Sharded Data Parallel (FSDP) [71], activation checkpointing, gradient accumulation, and sequence packing to minimize padding in the image-captioning dataset.

We assess model performance on three held-out data subsets: interleaved data (Obelics), image-caption data (HQITP), and text-only data (DCLM), following prior works [2, 3, 31]. This setup provides a robust evaluation of model generalization across diverse data types.

# B   Detailed extrapolation results

We report the detailed per-domain and per-model size MRE corresponding to each experiment and scaling law in the paper.

## B.1   Comparison to the laws of Ye et al. [68] and Liu et al. [40]

Ye et al. [68] propose four laws to model the behavior of the loss on a domain *as a function of $h$ only*, that is, for a fixed $N, D$ budget. They propose the following formulas, rewritten with notations consistent with our notation:

$$\mathcal{L}(h) = E + \sum_{i=1}^{k} C_i \exp(\gamma_i h_i) \qquad \text{(Ye M1)}$$

| Params | 275M | 468M | 932M | 1.63B | 2.28B | 3.35B | 8.13B |
|---|---|---|---|---|---|---|---|
| width | 800 | 1088 | 1632 | 2208 | 2624 | 3232 | 5120 |
| depth | | | | 24 | | | |
| Learning rate | 1.5e-3 | 1.5e-3 | 5e-4 | 4.2e-4 | 4e-4 | 3.5e-4 | 2.4e-4 |
| Training tokens | | | | 2.5B-600B | | | |
| Optimizer | | | | Fully decoupled AdamW [41] | | | |
| Optimizer Momentum | | | | $\beta_1 = 0.9, \beta_2 = 0.95$ | | | |
| Minimum Learning rate | | | | 0 | | | |
| Weight decay | | | | 1e-4 | | | |
| Batch size | | | | 2k | | | |
| Patch size | | | | (14, 14) | | | |
| Gradient clipping | | | | 1.0 | | | |
| Warmup iterations | | | | 1k | | | |
| Augmentations: | | | | | | | |
|   RandomResizedCrop | | | | | | | |
|     size | | | | 224px | | | |
|     scale | | | | [0.4, 1.0] | | | |
|   RandomHorizontalFlip | | | | $p = 0.5$ | | | |

Table 6: **Pre-training hyperparameters** used for pre-training of NMM to conduct the scaling laws study.

| Params | 90M | 200M | 350M | 700m | 1.3B | 3B |
|---|---|---|---|---|---|---|
| Width | 512 | 768 | 1024 | 1536 | 2048 | 3072 |
| Depth | | | | 24 | | |
| Learning rate | | | constant after warmup: 1e-4 | | | |
| Training tokens | 8.4B | 8.4B | 8.4B | 16.8B | 33.6B | 67.2B |
| Optimizer | | | AdamW [41] | | | |
| Optimizer Momentum | | | $\beta_1 = 0.9, \beta_2 = 0.95$ | | | |
| Batch size | | | 128 | | | |
| Sequence length | | | 1024 | | | |
| Gradient clipping | | | 1.0 | | | |
| Warmup iterations | | | 1k | | | |

Table 7: **Pre-training hyperparameters** used for the pre-training of LLM with PILE dataset to conduct the analyses

$$\mathcal{L}(h) = E + C \sum_{i=1}^{k} \exp(\gamma_i h_i) \tag{Ye M2}$$

$$\mathcal{L}(h) = E + C \exp(\prod_{i=1}^{k} \gamma_i h_i) \tag{Ye M3}$$

$$\mathcal{L}(h) = E + C \exp(\sum_{i=1}^{k} \gamma_i h_i) \tag{Ye M4}$$

where the parameters of the law are the $E, C, C_i, \gamma_i$. Liu et al. [40] propose to model the loss as a linear function of $h$. We compare this with the form of our scaling law eq. (4) when $N$ and $D$ are fixed:

$$\mathcal{L}(h) = E + \frac{1}{\sum_{i=1}^{k} C_i h_i^{\gamma_i}} \tag{Additive, fixed (N, D)}$$

We fit all those scaling laws on the LLM training data, keeping only one value for $N, D$ (we take $N = 200m$, $D = 8B$ tokens). We only keep 25 training mixtures to fit the laws, and report the MRE on the $84 - 25 = 59$ remaining in Tab. 13. Note that we had trouble fitting the M3 law, which is also reported to underperform in [68]. Overall, our formula gives systematically better training errors, and it most of the time translates to better testing errors on unseen mixtures. We stress once again that one of the core contributions of our work is to explain how scale interacts with data mixtures, by proposing scaling laws that take $N, D$, and $h$ as inputs. [68] only consider scaling laws with respect to $h$.

| Scaling Law | Domain | Train MRE(%) | 3B MRE(%) |
|---|---|---|---|
| Simple | Arxiv | 0.55 | 2.40 |
| Additive | Arxiv | 0.50 | 2.09 |
| Joint | Arxiv | 0.39 | 1.62 |
| Full | Arxiv | 0.39 | 1.83 |
| Simple | Book | 0.38 | 1.11 |
| Additive | Book | 0.29 | 0.80 |
| Joint | Book | 0.24 | 1.19 |
| Full | Book | 0.24 | 1.14 |
| Simple | C4 | 0.35 | 0.47 |
| Additive | C4 | 0.29 | 0.31 |
| Joint | C4 | 0.24 | 0.34 |
| Full | C4 | 0.23 | 0.41 |
| Simple | GitHub | 0.81 | 2.05 |
| Additive | GitHub | 0.65 | 1.17 |
| Joint | GitHub | 0.54 | 2.51 |
| Full | GitHub | 0.52 | 1.97 |
| Simple | Commoncrawl | 0.34 | 0.65 |
| Additive | Commoncrawl | 0.29 | 0.58 |
| Joint | Commoncrawl | 0.24 | 0.90 |
| Full | Commoncrawl | 0.23 | 0.74 |
| Simple | Stackexchange | 0.57 | 0.68 |
| Additive | Stackexchange | 0.51 | 0.36 |
| Joint | Stackexchange | 0.38 | 0.47 |
| Full | Stackexchange | 0.36 | 0.42 |
| Simple | Wikipedia | 0.97 | 4.53 |
| Additive | Wikipedia | 0.92 | 4.45 |
| Joint | Wikipedia | 0.57 | 2.09 |
| Full | Wikipedia | 0.56 | 1.98 |

Table 8: Full results of experiments in Sec. 4 for the LLM experiment.

| Scaling Law | Domain | Train MRE(%) | 2B MRE(%) | 8B MRE(%) |
|---|---|---|---|---|
| Simple | Text | 0.15 | 0.44 | 0.50 |
| Additive | Text | 0.12 | 0.40 | 0.51 |
| Joint | Text | 0.10 | 0.39 | 0.32 |
| Full | Text | 0.09 | 0.38 | 0.33 |
| Simple | Image-Captions | 0.52 | 0.89 | 1.36 |
| Additive | Image-Captions | 0.47 | 0.83 | 1.23 |
| Joint | Image-Captions | 0.43 | 0.85 | 1.17 |
| Full | Image-Captions | 0.43 | 0.90 | 1.33 |
| Simple | Interleaved | 0.22 | 0.65 | 0.80 |
| Additive | Interleaved | 0.14 | 0.44 | 0.58 |
| Joint | Interleaved | 0.10 | 0.41 | 0.45 |
| Full | Interleaved | 0.10 | 0.40 | 0.45 |

Table 9: Full results of experiments in Sec. 4 for the multimodal experiment.

| Scaling Law | Domain | Train MRE(%) | 1B MRE(%) |
|---|---|---|---|
| Simple | Noisy image-text | 0.34 | 1.05 |
| Additive | Noisy image-text | 0.35 | 1.20 |
| Joint | Noisy image-text | 0.23 | 0.63 |
| Full | Noisy image-text | 0.21 | 0.58 |
| Simple | Synthetic | 1.85 | 5.96 |
| Additive | Synthetic | 1.89 | 6.19 |
| Joint | Synthetic | 0.83 | 5.94 |
| Full | Synthetic | 0.70 | 5.56 |
| Simple | High quality 1 | 0.70 | 0.99 |
| Additive | High quality 1 | 1.19 | 2.02 |
| Joint | High quality 1 | 0.34 | 1.18 |
| Full | High quality 1 | 0.32 | 1.08 |
| Simple | High quality 2 | 0.64 | 1.22 |
| Additive | High quality 2 | 0.64 | 0.79 |
| Joint | High quality 2 | 0.31 | 1.06 |
| Full | High quality 2 | 0.31 | 0.93 |

Table 10: Full results of experiments in Sec. 4 for the LVM experiment.

| Scaling Law | Domain | Train MRE(%) | 700m MRE(%) | 1B MRE(%) |
|---|---|---|---|---|
| Simple | Wikipedia | 0.28 | 0.75 | 1.13 |
| Additive | Wikipedia | 0.24 | 0.77 | 1.11 |
| Joint | Wikipedia | 0.13 | 0.24 | 0.39 |
| Full | Wikipedia | 0.12 | 0.23 | 0.39 |
| Simple | GitHub | 0.60 | 1.38 | 3.28 |
| Additive | GitHub | 0.42 | 1.10 | 1.69 |
| Joint | GitHub | 0.23 | 0.38 | 1.46 |
| Full | GitHub | 0.22 | 0.49 | 1.91 |
| Simple | StackExchange | 0.40 | 0.90 | 1.50 |
| Additive | StackExchange | 0.33 | 0.88 | 1.31 |
| Joint | StackExchange | 0.17 | 0.26 | 1.05 |
| Full | StackExchange | 0.16 | 0.30 | 1.17 |
| Simple | PG-19 | 0.21 | 0.53 | 0.91 |
| Additive | PG-19 | 0.16 | 0.55 | 0.94 |
| Joint | PG-19 | 0.15 | 0.40 | 0.71 |
| Full | PG-19 | 0.14 | 0.35 | 0.54 |

Table 11: Full results of experiments in Sec. 6 for the LLM experiment.

| Scaling Law | Domain | Train MRE(%) | 700m MRE(%) | 1B MRE(%) |
|---|---|---|---|---|
| Additive | Wikipedia | 0.45 | 0.53 | 0.53 |
| Joint | Wikipedia | 0.22 | 0.18 | 0.19 |
| Additive | GitHub | 0.70 | 0.88 | 1.49 |
| Joint | GitHub | 0.39 | 0.31 | 1.09 |
| Additive | StackExchange | 0.50 | 0.55 | 0.50 |
| Joint | StackExchange | 0.29 | 0.23 | 0.44 |
| Additive | PG-19 | 0.24 | 0.37 | 0.53 |
| Joint | PG-19 | 0.18 | 0.25 | 0.44 |

Table 12: Full results of experiments in Sec. 4 for the cosine schedule experiment.

| Scaling law | Domain | Train (MRE%) | Test (MRE%) |
|---|---|---|---|
| Additive, fixed (N, D) | Wikipedia | **0.07** | 0.18 |
| Ye M1 | Wikipedia | 0.08 | **0.14** |
| Ye M2 | Wikipedia | 0.09 | 0.17 |
| Ye M3 | Wikipedia | 2.54 | 4.20 |
| Ye M4 | Wikipedia | 0.17 | 0.31 |
| Regmix | Wikipedia | 0.92 | 1.31 |
| Additive, fixed (N, D) | GitHub | **0.10** | **0.19** |
| Ye M1 | GitHub | 0.20 | 0.61 |
| Ye M2 | GitHub | 0.22 | 0.40 |
| Ye M3 | GitHub | 5.45 | 5.26 |
| Ye M4 | GitHub | 0.36 | 0.44 |
| Regmix | GitHub | 0.65 | 1.35 |
| Additive, fixed (N, D) | StackExchange | **0.07** | **0.18** |
| Ye M1 | StackExchange | 0.14 | 0.32 |
| Ye M2 | StackExchange | 0.14 | 0.21 |
| Ye M3 | StackExchange | 4.11 | 3.20 |
| Ye M4 | StackExchange | 0.22 | 0.34 |
| Regmix | StackExchange | 0.78 | 0.92 |
| Additive, fixed (N, D) | PG-19 | **0.08** | **0.12** |
| Ye M1 | PG-19 | 0.09 | 0.17 |
| Ye M2 | PG-19 | 0.13 | 0.18 |
| Ye M3 | PG-19 | 2.21 | 3.14 |
| Ye M4 | PG-19 | 0.16 | 0.21 |
| Regmix | PG-19 | 0.64 | 0.89 |

Table 13: Comparison of our scaling laws for a fixed (N, D) budget with those of [68] and [40] on the LLM data.

## B.2 Comparison to Ge et al. [23]

Ge et al. [23] propose a scaling law to evaluate the loss the $i-th$ training domains, as a function of both $h$ and number of tokens $D$:

$$\mathcal{L}_i(h, D) = \left( \frac{B}{D^\beta} + E \right) \frac{C}{h_i^\gamma} \qquad \text{(Ge 24)}$$

where the coefficients $B, \beta, E, C$ and $\gamma$ have to be fitted. Here, the loss must be on domain $i$, and it only involves the proportion of that domain $h_i$, not that of the other domains. In our view, this is a caveat since it implies that all other domains would have the same impact on the loss for domain $i$, while one other training domain might be very useful for that task, and another might be useless.

Since this law does not take into account model scale, we compare it to our additive law for a fixed model scale:

$$\mathcal{L}(h, D) = E + \frac{1}{\sum_{i=1}^{k} C_i h_i^{\gamma_i}} + \frac{B}{D^\beta} \qquad \text{(Additive, fixed N)}$$

We consider a fixed size of model (N=200M) on the LLM training experiment, take only 25 mixtures to fit the scaling laws, and report the MRE on the remaining testing set in Tab. 14. We see that our proposed scaling law achieves a significantly lower MRE on both train and testing data, highlighting the importance of taking all other domains into account.

| Scaling law | Domain | Train (MRE%) | Test (MRE%) |
|---|---|---|---|
| Additive, fixed N | Wikipedia | **0.13** | **0.17** |
| Ge 24 | Wikipedia | 0.51 | 0.62 |
| Additive, fixed N | GitHub | **0.20** | **0.26** |
| Ge 24 | GitHub | 1.99 | 2.26 |
| Additive, fixed N | StackExchange | **0.14** | **0.19** |
| Ge 24 | StackExchange | 0.94 | 1.21 |
| Additive, fixed N | PG-19 | **0.13** | **0.15** |
| Ge 24 | PG-19 | 0.49 | 0.54 |

Table 14: Comparison of our scaling laws for a fixed model size $N$ with that of [23] on the LLM data.

## C   Additional analysis

### C.1   Cosine learning rate scheduler.

The bulk of our experiments use a constant learning rate, which helps us gather many $D$ points for each run, but this departs from what is done in practice when training competitive models, where a cosine learning rate is typically used. In order to validate that our scaling laws are still valid when training with a cosine learning rate, we repeat the LLM experiments with $k = 4$ pile domains with cosine learning rate decays, with fewer runs, training for $5$ different values of $D$, with $25$ different domain weights. We use $90M, 200M, 350M$ and $700M$ models for training, and extrapolate to $1.3B$. We observe that our scaling law fit is similar to those in the main experiments: on the 1B model, we get to an average MRE of $0.76\%$ for the additive law and $0.54\%$ for the joint law. We report detailed results in appendix B. Interestingly, the estimated optimal domain weights for the average loss are very similar to those estimated with the constant learning rate: for the additive law, we have $h^*_{\cos} = [0.35, 0.18, 0.30, 0.17]$ and $h^*_{\text{const}} = [0.34, 0.17, 0.32, 0.17]$.

### C.2   Asymptotic behavior

We can get an information-theoretic explanation of the bias term in the scaling laws. Let $p$ be the true data distribution of the target domain. Let $q(h)$ be the output distribution of a model of size $N \to \infty$ trained for $D \to \infty$ tokens with domain weights $h$. Let $h^*$ be the optimal domain weights, which minimizes the scaling law $\mathcal{L}(+\infty, +\infty, h) = E + (\sum_{i=1}^{k} C_i h_i^{\gamma_i})^{-1}$, i.e the cross-entropy term $CE(p, q(h)) = H(p) + KL(p\|q(h))$, with $H(\cdot)$ the Shannon entropy and $KL(\cdot\|\cdot)$ the Kullback-Leibler divergence. We have the following decomposition:

$$CE(p, q(h)) = \underbrace{H(p) + KL(p\|q(h^*))}_{\text{constant, independant of h}} + \underbrace{KL(p\|q(h)) - KL(p\|q(h^*))}_{\geq 0 \text{ by hypothesis on } h^*}$$

$$= E + (\sum_{i=1}^{k} C_i h_i^{*\gamma_i})^{-1} + (\sum_{i=1}^{k} C_i h_i^{\gamma_i})^{-1} - (\sum_{i=1}^{k} C_i h_i^{*\gamma_i})^{-1}.$$

(11)

We can identify both terms since they are of the form "constant plus non-negative function that cancels". We see that $E + (\sum_{i=1}^{k} C_i h_i^{*\gamma_i})^{-1}$ captures both the intrinsic entropy $H(p)$ of the target distribution, and the shift $KL(p\|q(h^*))$ induced by training on the optimal mixture $h^*$, while the right-hand term is the expected log-likelihood ratio $\mathbb{E}_p[\log \frac{q(h^*)}{q(h)}]$, which measures how far the model trained on $h$ is from the optimal one. If $p$ is one of the training domains $\mathcal{D}_i$, for disjoint domains we can assume $h_i^* \approx 1$ (see next section for justification) and simply bound its entropy $H(p) \leq E + C_i^{-1}$.

### C.3   Optimal domain weights

**Optimal weights for the additive scaling law.**   We report the optimal domain weights $h^*$ for the additive scaling law in Tab. 15.

Recall that  Sec. 5 defines the optimal domain weights $h^*(\cdot)$ as function of compute budget $(N, D)$. In the main text, we assumed uniform weights of the target domains $\mathcal{D}_i$. However, we can also

Table 15: Optimal domain weights $h^*$ for LLMs. Note that for the OpenHermes optimal weights, the law predicted a weight of 0 for wikipedia, which we artificially set to 1%.

| Model | Arxiv | Book | C4 | GitHub | Commoncrawl | Stackexchange | Wikipedia |
|---|---|---|---|---|---|---|---|
| For average loss $h^*_{avg}$ | 9.6 | 9.5 | 25.1 | 8.0 | 12.1 | 17.9 | 17.0 |
| For OpenHermes $h^*_{OH}$ | 9.4 | 4.8 | 27.8 | 6.6 | 36.9 | 13.5 | 1.0 |
| Base | 4.6 | 4.2 | 26.7 | 5.2 | 52.2 | 3.3 | 3.8 |

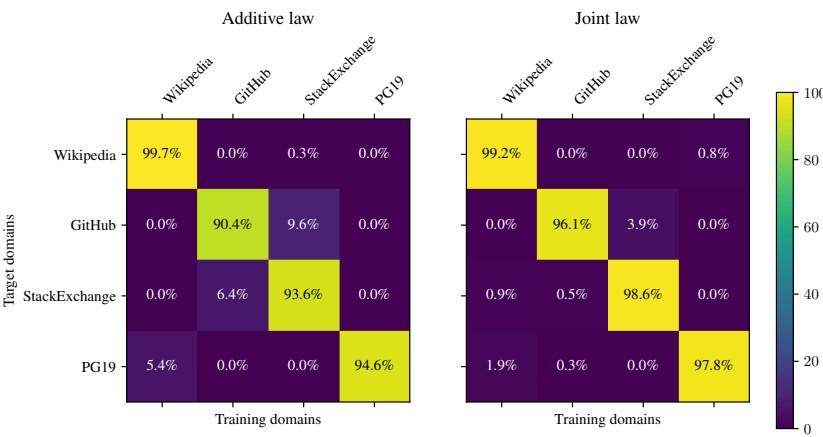

Figure 7: **Optimal domain weights for a single (pure) domain, typically lies at the corner of the probability simplex.** Scaling law predictions for a 1.3B model trained on 10B tokens.

consider a weighted average scenario, with an arbitrary weight vector $w_i$.

$$h^*(w, N, D) \in \arg \min_{h \in \Delta_k} \sum_{i=1}^{m} w_i \mathcal{L}^i(N, D, h). \tag{12}$$

Once again, this objective is seamlessly optimizable with mirror descent. When training domains and target domains are the same, $w$ and $h^*$ are a probability distribution over the same simplex. Therefore, we can study the mapping $w \mapsto h^*(w, N, D)$.

**Behavior at the corners**  We predict the optimal domain weights for both laws, chosing each training domain $j$ as the target, that is, putting $w_i = 1$ if $i = j$ and $w_i = 0$ otherwise. The results are given in Fig. 7.

We see that the data mixture law predicts that the optimal domain weights are typically located in the corresponding corner of the simplex - which is not too surprising when there is little domain overlap. This justifies the rule of thumb $h^*_i \approx 1$ when the target domain is $\mathcal{D}_i$.

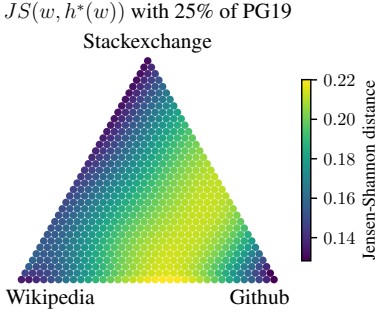

Figure 8: **The optimal domain weights are always different from the target mixture, except at the corners of the simplex** 1.3B model with 10B tokens, on 4 domains of The Pile.

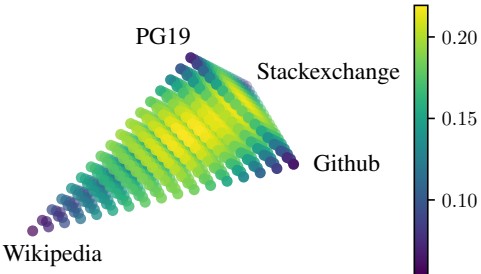

Jensen-Shannon metric $JS(w, h^*(w))$

Figure 9: **Jensen-Shannon distance between target mixture** $h$ **and its optimum training mixture** $h^*(w)$ on the 4D simplex. 1.3B model with 10B tokens, on 4 domains of the Pile. No fixed-point are visible, except at the corners. This suggests that in general, it is better not to train on the mixture you want to be good on.

**Fixed-points** For a given $(N, D)$ pair, the optimization problem of eq. (12) defines a function $w \mapsto h^*(w)$ that maps the simplex onto itself. The fact that $h^*(w) \neq w$ indicates the surprising phenomenon that, in order to minimize the loss $\mathcal{L}^{\text{ERM}} = \sum_{i=1}^{m} w_i \mathcal{L}^i(\theta)$, it is faster to instead minimize $\mathcal{L}^* = \sum_{i=1}^{m} h^*(w)_i \mathcal{L}^i(\theta)$, rather than minimizing directly $\mathcal{L}^{\text{ERM}}$.

Fixed points of the map $w \mapsto h^*(w)$ correspond to target mixtures $w$ that are minimized by training on $w$ itself: this is the empirical risk minimizer.

To find these points, we compute the Jensen-Shannon distance, defined as $JS(w, h^*) = \sqrt{1/2(KL(w\|m) + KL(h^*\|m))}$ with $m = (w + h^*)/2$, and we look for near-zero values, in Fig. 8 and Fig. 9.

**Accumulation point of asymptotes.** In mirror of Fig. 6 we can monitore how the optimal mixture $h^*$ evolves as we scale parameters $N$ and data $D$. For example, we can keep $D$ constant and scale $N$, or the opposite, or scale both simultaneously with $D \propto N$. Results are highlighted in Figure 10. These asymptotes reach an accumulation point when $D \to \infty$ or $N \to \infty$. They depend on the speed at which $N$ and $D$ grow.

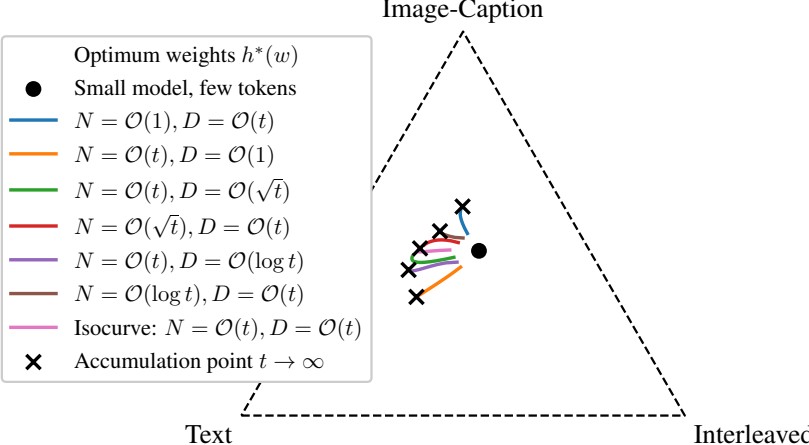

Figure 10: **Asymptotes of optimal mixture when increasing $N$ and $D$ at different speeds** on multimodal data. Surprisingly, there is little diffenrence between proportional scaling $\mathcal{O}(t)$ and square-root scaling $\mathcal{O}(\sqrt{t})$: both are fast enough. However, when one quantity is held constant, or only grow at logarithmic speed, the accumulation point changes.

