# OpenReview forum: "Scaling Laws for Optimal Data Mixtures"
_NeurIPS.cc/2025/Conference — NeurIPS 2025 poster_

### Official Review · Reviewer_QVnK · 2025-06-30

**Clarity:** 4
**Significance:** 4
**Originality:** 2
**Rating:** 5
**Confidence:** 4

**Summary:**

The paper proposes the use of scaling laws to select data mixtures for pretraining language and multimodal models. They adapt the Chinchilla scaling laws to account for a vector of domain weights (= proportion of each domain/source in the data mixture, that sums to 1). They propose the following method to select optimal data mixtures:

1. Train a set of small models (evenly spaced N and D) with different domain weight vectors (also evenly spaced on a grid).
2. Minimize the selected loss function (on source or target domain) to fit scaling laws.
3. Select the data mixture with the best predicted (extrapolated) loss value.

They demonstrate that their method holds for new data mixtures and domain weights.

**Questions:**

1. How are the hyperparameters (eg. LR, batch size) set? Are they the same for all model sizes, or do you spend compute finding optimal HPs for each model size? Will the results hold with different HPs?
2. You mention that the model sizes are varied by changing only the width dimension. Did you try varying num_layers at all? I’ll be curious to know if that significantly impacts the error rates.
3. The GPU hours that you’ve listed - do they include the target model sizes on which the method is validated?
4. You mention that only 10-20 runs are needed for fitting scaling laws. How much compute is that in terms of % of target model? It will be helpful to highlight that as well.
5. It looks like the number of runs required tends to scale with the number of domains. Is that depended on the size of each domain, as well? Eg. if the domain is too small / fine-grained, do the results still hold? More generally, is there a minimum threshold for the domain size?
6. Re: cosine LR scheduler, it’s very interesting that the optimal domain weights seem to be similar to those estimated with constant LR. Do you think it’s possible to fit scaling laws based on models trained with constant LR scheduler, and directly extrapolate the perf of a target model trained with cosine LR?

**Ethical Concerns:**

["NO or VERY MINOR ethics concerns only"]

**Final Justification:**

No change to my positive rating. The paper presents a method that is practically useful for determining optimal data mixtures. My one comment on using 8B model has been addressed.

**Limitations:**

Yes

**Paper Formatting Concerns:**

Lines 163-165: Capitalization is inconsistent. Eg. “Github” and “GitHub”, “StackExchange” vs “Stackexchange”, “pg19” etc.

**Quality:**

4

**Strengths And Weaknesses:**

Strengths:

1. Very clearly articulated problem, very relevant for pretraining. The experimental setup is quite thorough - including multimodal as well as language models. Results are also very well organized.
2. I particularly appreciate that the authors have explicitly listed the GPU hours used for training the models.

Weaknesses:

1. The largest model size that was validated is 8B. While I appreciate that these experiments are expensive to run, language models in particular exhibit “lift-off” on certain capabilities beyond that scale, which can cause a shift in the fitted scaling laws, depending on what’s used as the target loss function. It will be helpful to include some discussion of this in the paper.

---

> ### Author Rebuttal · Authors · 2025-07-30
>
> We thank the reviewer for these excellent points and questions. Below we address each in turn:
>
>
> ### On Extrapolation Beyond 8B and Emergent Capabilities
>
> We agree that some language model capabilities emerge beyond certain parameter thresholds, which could cause a shift in performance trajectories. However, these emergent behaviors typically manifest in downstream task metrics, while the training loss--which we use as the target for our scaling laws--remains smooth and well-behaved. That said, we acknowledge that extrapolation to extreme large model scales  remains challenging. We will expand the discussion in the paper to make this limitation explicit.
>
>
> ### Hyperparameter Selection Across Model Sizes
>
> Our hyperparameters are consistent across scales and follow smooth trends with model size. For example, learning rate scales as 1/dim for LLMs and follows a power law found empirically for multimodal models. These choices are found based on our previous experience training these models and are detailed in Appendix A.
>
>
> ### Width vs. Depth Scaling
>
> As noted, we vary the model size by changing width, holding the number of layers constant. We did not experiment with varying depth in this study, but agree that it would be a worthwhile direction for future work.
>
>
> ### GPU Hours and Target Model Inclusion
>
> Yes, the GPU hours we report include both the runs used to fit the scaling law and the larger target model runs used for validation. We will clarify this in the revised version.
>
>
>
> ### Compute Overhead of Scaling Law Fitting:
>
> As noted, 10–20 runs are typically sufficient for fitting the law. In terms of compute, we will also improve the paper by disentangling the flops required to train models to fit the law and those required for the large scale runs.
>
> We use the following assumptions: Flops are computed as 6ND, and we use a number of training runs that is enough to get to a low MRE, as per Fig.4. We report the training costs for the model sizes shown in table 1. NMM (using 6 runs per sizes): small scale runs cost: 6.9e21 flops. Cost of one 8B run: 7.8e21 flops (Note that we could have trained this model for longer, linearly increasing the computational costs. Also, reducing the number of training tokens to 60B for the small scale runs does not significantly impact the eval MRE, while it reduces the small scale flops to 4.1e21). LLM (using 10 runs per size): small scale runs cost: 2.9e20 flops. Cost of one 3B run: 1.1e21 flops, which is approximately five times higher. We will add these important numbers in the final version of the paper.
>
> ### Impact of Number and Size of Domains
>
> The number of required runs increases with the number of domains, since the number of coefficients to fit grows accordingly. This is because our formulation treats each domain as a separate component in the mixture vector h. In this work, we assume non-repeating data, where each training step samples from the domains without reuse. We’ll make this assumption clearer in the paper.
> The case of repeating data is a very interesting future research direction, which becomes more and more pressing as data is being repeated more often in standard pre-training pipelines.
>
>
> ### Transferability Across Learning Rate Schedulers
>
> Yes, we tested this. We have run a new, large scale LLM experiment.
> The study is as follows:
> We train llms on 7 domains from slimpajama, and fit the scaling laws with small scale runs (<1B parameters). We then train large 7b models with 160B tokens with the optimal domain weights to get the best loss on average for the 7 domains (model A) , or to get the best loss on OpenHermes, a high quality domain (model B). We compare these models to models trained on the base distribution (Model C) for slimpajama and the uniform distribution (Model D) across the 7 domains. To get the best performing models, we train them with a cosine schedule, while the small scale runs were done with constant lrs. In terms of loss, Model A achieves the best loss on average over the domains, and model B achieves the best loss on openhermes. This shows that the optimal weights found by our laws still yield superior performances at large scales. We then score those four 7B models on standard benchmarks. Model B, targeting a high quality domain, yields the best scores, followed by model A, while D and C are comparable. This demonstrates that the optimal weights found at smaller scale to optimize the next token prediction loss also yield good performances on downstream tasks. The downstream tasks' results are as follows (CORE is the median of low variance benchmarks (Li et al. 2024)):
>
> |Model | CORE median score| MMLU|
> | -------- | ------- |--------|
> |Optimal weights for avg| 56% | 32% |
> |Optimal weights for OpenHermes|  **58%** |**37%**|
> |Base | 52% | 25% |
> |Uniform | 53% | 30% |
>
>
> In that large scale experiment, we empirically found that the optimal domain weights estimated using models trained with a constant learning rate transfer well to models trained with a cosine schedule.
>
> However, we do not believe it is possible to *extrapolate* the loss of the model trained with the cosine schedule directly, since this model would typically have a much lower loss than the same model trained with a constant learning rate. We believe that extrapolating the loss would require having at least a few small scale runs with a cosine schedule. How to best incorporate the fixed lr runs to these small scale cosine runs to get a robust scaling law estimation is a very interesting research direction for future work.
>
>
> We thank you again for your review, and we hope that we have clarified any grey areas.
>
>
> *Li, Jeffrey, et al. "Datacomp-lm: In search of the next generation of training sets for language models." Advances in Neural Information Processing Systems 37 (2024): 14200-14282.*

---

### Official Review · Reviewer_mQkL · 2025-07-02

**Clarity:** 2
**Significance:** 2
**Originality:** 2
**Rating:** 3
**Confidence:** 5

**Summary:**

This paper examines the prediction of model loss as a function of data size, model size, and domain weights. The paper proposes several parametric functions to model this relationship and evaluates their prediction accuracy in the pre-training of LLMs and multimodal models.

**Questions:**

1. What is the computational cost required for fitting the parameters of the scaling law? How does it compare to the cost of training the model at the target scale?

2. Is there any intuition why cosine learning rate and constant learning rate yield similar domain weights?

**Ethical Concerns:**

["NO or VERY MINOR ethics concerns only"]

**Final Justification:**

The paper, motivated by the problem of data mixing, has not yet been evaluated comprehensively in terms of identifying an effective data mixture. Specifically, many key baselines are missing. While the authors insist that their contributions also include finding a scaling law of loss as a function of data, compute, and mixing ratio, it is only a building block towards identifying an optimal mixture, which is the main motivation as stated in the abstract. Overall, the paper needs to rigorously evaluate against these baselines using already established metrics, such as downstream performance yielded by the data mixture, wihtout which the contribution is not convincing enough.

**Limitations:**

Yes.

**Paper Formatting Concerns:**

None.

**Quality:**

2

**Strengths And Weaknesses:**

Strengths:

1. The paper examines an important and challenging topic, particularly given the substantial computational costs involved in performing the experiments.
2. The paper evaluates both LLMs and multi-modal models, a scope that is rarely explored in the literature.

Weaknesses:
1. As discussed in the related work section, there are several papers that share a similar goal of modeling the model loss and/or identifying the optimal mixture (such as DoReMi, Regmix, Data mixing law, and Doge). However, this paper lacks an empirical comparison with these papers. In particular, the paper overlooks a comparison with AutoScale [1], which addresses the same problem and appears to be more computationally efficient in scaling law fitting since their scaling law is more parameter-efficient.
2. Related to the first weakness, the experiment's empirical evaluation is limited. The authors do not clearly state the evaluation metrics and appear to rely solely on the convergence of training loss to assess a mixture's effectiveness. A comprehensive evaluation would include validation loss and performance on relevant downstream benchmarks, which are of greater interest to practitioners.
3. The paper's proposed scaling law as a function of mixing ratio lacks theoretical justification or intuition for its design.

[1] Kang et al. AutoScale: Scale-Aware Data Mixing for Pre-Training LLMs. 2024.

---

> ### Author Rebuttal · Authors · 2025-07-30
>
> Dear reviewer,
>
> We thank you for your helpful feedback, which will help us improve the paper. The paper Autoscale that you mention is indeed relevant, and we will make sure to include a comparison in the final version. We now reply to your questions.
>
> ### Comparison with Related Work (AutoScale, DoReMi, RegMix, Doge)
>
> We want to highlight that the primary goal of our paper is to give a way to model the loss as a function of (N, D, h). None of the works that you mention do this: Autoscale models the loss as a function of (D, h) but not N, and Regmix only as a function of (h). DoreMi and Doge **do not** model the loss.
>
> One of the applications - but not the only contribution - of our work is to estimate the optimal data mixture as a function of (N, D), which is indeed a goal shared by all the works you mention. However, DoreMi and Doge respectively rely on costly Distributionally robust optimization and bilevel optimization to estimate the optimal mixture, which incurs a significant change to the training loop and a very large compute/memory overhead. On the other hand, our work, Autoscale and Regmix only observe the loss as a function of N, D, h, which does not require any change to the training loop. We therefore defer a comparison to Doremi and Doge to future work.
>
> Upon your remark, we have added a comparison to Regmix and Autoscale, in the same setup as in appendix B.
>
> For regmix, in the fixed (N, D) case, we get the following test MREs % :
>
>
> | Scaling law formula | Wikipedia | GitHub|Stackexchange|Pg-19|
> |------|------|------|------|------|
> | Additive, fixed (N, D) | **0.18**| **0.19**|**0.18**|**0.12**|
> | Regmix | 1.31|1.35|0.92|0.89|
>
>
> While for autoscale, in the fixed (N) case, we get the following test MREs %:
>
> | Scaling law formula | Wikipedia | GitHub|Stackexchange|Pg-19|
> |------|------|------|------|------|
> | Additive, fixed (N) | **0.17**| **0.26**|**0.19**|**0.15**|
> | Autoscale | 1.94 |2.03|1.31|1.18|
>
> The MREs obtained with our proposed scaling laws are far lower than those of Regmix or Autoscale.
>
> We will add this insightful experiment to the text. This comparatively higher MRE for AutoScale means that the estimated optimal domain weights with that method can be far from optimality.
> The second part of AutoScale consists of a method to extrapolate from optimal domain weights at a small scale to a large scale. Since our scaling laws directly predict the loss as a function of (N, D, h), we can directly estimate h^*(N, D), as explained in Sec. 5, while AutoScale requires a two-stage procedure.
>
> We thank you again for recommending the inclusion of AutoScale as a comparison. This will help better situate our contribution in comparison to prior work and provide an improved sense of the utility of our proposed method.
>
> ### Evaluation Metrics and Empirical Scope:
>
>
> Our primary evaluation metric is Mean Relative Error (MRE) on held-out loss predictions, which is a standard measure for assessing the accuracy of scaling law fits.
>
> We want to clarify a misunderstanding here, which we will specifically highlight in the final version of the paper: **we never report training losses** (since the training loss of a run depends on the training mixture). Instead, all the losses that we report in the paper are validation losses, computed on a fixed held-out subset of the domain's data.
>
>
> Additionally, Figure 1 includes validation losses, which show that standard uniform mixtures perform worse than those selected via our method.
>
>
> We agree that downstream performance is ultimately what matters. To address this, upon your remarks, we added a new study with larger-scale LLMs (up to 7B parameters), where we show that the model trained with our predicted optimal mixture achieves the best results.
> The study is as follows:
> We train llms on 7 domains from slimpajama, and fit the scaling laws with small scale runs (<1B parameters). We then train large 7b models with 160B tokens with the optimal domain weights to get the best loss on average for the 7 domains (model A) , or to get the best loss on OpenHermes, a high quality domain (model B). We compare these models to models trained on the base distribution (Model C) for slimpajama and the uniform distribution (Model D) across the 7 domains. To get the best performing models, we train them with a cosine schedule, while the small scale runs were done with constant lrs. In terms of loss, Model A achieves the best loss on average over the domains, and model B achieves the best loss on openhermes. This shows that the optimal weights found by our laws still yield superior performances at large scales. We then score those four 7B models on standard benchmarks. Model B, targeting a high quality domain, yields the best scores, followed by model A, while D and C are comparable. This demonstrates that the optimal weights found at smaller scale to optimize the next token prediction loss also yield good performances on downstream tasks. The downstream tasks' results are as follows (CORE is the median of low variance benchmarks (Li et al 2024)):
>
> |Model | CORE median score| MMLU|
> | -------- | ------- |--------|
> |Optimal weights for avg| 56% | 32% |
> |Optimal weights for OpenHermes|  **58%** |**37%**|
> |Base | 52% | 25% |
> |Uniform | 53% | 30% |
>
> Therefore, the optimal weights found with our method also yield models with superior downstream performances.
>
> *Li, Jeffrey, et al. "Datacomp-lm: In search of the next generation of training sets for language models." Advances in Neural Information Processing Systems 37 (2024): 14200-14282.*
>
>
> ### Lack of Theoretical Justification for the Scaling Law Form
>
> While we do not include a thorough theoretical justification, we tried to explain our motivations and insights to obtain these laws (Sec. 2.2). Our work is mostly empirically driven: we experimented with several candidate formulations and selected the one that consistently minimized MRE across diverse settings. A theoretical characterization would be extremely interesting and help give a clearer interpretation for the components of the proposed scaling laws, and would be valuable future work. However, this is not necessary; demonstrable empirical performance is sufficient for establishing practical pretraining pipelines.
>
>
>
> ### Question 1
>
> This is an interesting point. We use the following assumptions: Flops are computed as 6ND, and we use a number of training runs that is enough to get to a low MRE, as per Fig.4. We report the training costs for the model sizes shown in table 1. NMM (using 6 runs per sizes): small scale runs cost: 6.9e21 flops. Cost of one 8B run: 7.8e21 flops (Note that we could have trained this model for longer, linearly increasing the computational costs. Also, reducing the number of training tokens to 60B for the small scale runs does not significantly impact the eval MRE, while it reduces the small scale flops to 4.1e21). LLM (using 10 runs per size): small scale runs cost: 2.9e20 flops. Cost of one 3B run: 1.1e21 flops, which is approximately five times higher. We will add these important numbers in the final version of the paper.
>
>
> ### Question 2
>
> We experimented with different learning rates schedulers to investigate if they have an effect on the laws. The results suggest that the lr schedulers are orthogonal to data mixture selection. The intuition is that the optimal domain weights are more about data quality / similarity, which is an intrinsic property of the data domains only, not of the architectural hyperparameters of the models that are trained.
>
>
> We thank you again for your review, and we hope that we have clarified any grey areas.

---

> > ### Comment · Reviewer_mQkL · 2025-08-08
> > **Thanks for the detailed rebuttal**
> >
> > I want to thank the authors for the detailed rebuttal.
> >
> > > Regarding the comparison with related work.
> >
> > As the authors note, the unified problem that all of these works, including the current paper (as suggested by the title), is to identify the optimal data mixture. The critical comparison should be on the downstream performance and validation loss of the final data mixture identified by each method.
> >
> > > Regarding the evaluation metrics
> >
> > It is great to see the addition of evaluations on downstream benchmarks, which is essential for this work. However, following the point above, RegMix and AutoScale should be included as baselines in the downstream evaluations. Comparing only against a uniform mixture is insufficient to show the effectiveness of the proposed method in identifying an optimal data mixture relative to prior literature.
> >
> > > Regarding the computational costs
> >
> > Thanks for providing the results. It will be good to add into the paper so that practitioners to understand the trade-offs and decide if the performance gains from this data curation method justify the additional computational expense. In particular, this paper's scaling law has more parameters than RegMix and AutoScale. The improved scaling prediction comes at the cost of computational overhead of getting the necessary data for fitting more parameters.
> >
> > > Regarding learning rate
> >
> > To substantiate the claim regarding the learning rate, it would be beneficial to include an evaluation using a different scheduler.

---

> ### Author Response · Authors · 2025-08-08
>
> Dear reviewer,
>
> We thank you for your reply. Your suggestions will help us improve the paper.
>
> However, we believe that the criticisms you have raised only concern small parts of the paper, and that your recommendation for rejection does not take our entire contribution into account. We are the first paper to propose a joint scaling law for (N, D, h); we carefully demonstrate that our laws extrapolate to a wide variety of domains, and we demonstrate that the optimal weights found by our laws lead to strong models, at a larger scale than most data mixing papers. In our view, these new findings are interesting for the LLM community.
>
> We now answer the questions you have raised.
>
> |  the unified problem that all of these works, including the current paper (as suggested by the title), is to identify the optimal data mixture
>
> We insist that, while this is indeed one of the key contributions of that paper, it is not the only one. This is the first work that predicts the loss of a model as a function of N, D and h, and which does so across an array of different modalities/domains.
>
> | It is great to see the addition of evaluations on downstream benchmarks, which is essential for this work.
>
> Thanks; we also think that this strengthens the paper significantly.
>
> | RegMix and AutoScale should be included as baselines in the downstream evaluations.
>
> We respectfully disagree with that point. Regmix and AutoScale are both based on the premise that they can accurately predict, for a fixed model size, the value of the loss on a domain from the number of tokens N and the mixture h. As seen in the table in our rebuttal above, these methods have a **test MRE that is significantly higher** than our method.
>
> We believe that it is therefore sufficient to conclude that our method improves upon them. Since RegMix and Autoscale cannot accurately capture the effect of h on the loss, we do not see how their estimated optimal weights could outperform the ones we find. It would indeed be interesting to quantify the extent of that difference, but we leave this for future work.
>
> |  It will be good to add into the paper so that practitioners to understand the trade-offs and decide if the performance gains from this data curation method justify the additional computational expense.
>
> Indeed, we believe that this improves the paper and makes it clearer.
>
> | In particular, this paper's scaling law has more parameters than RegMix and AutoScale. The improved scaling prediction comes at the cost of computational overhead of getting the necessary data for fitting more parameters.
>
> We respectfully disagree. Indeed, Regmix and Autoscale have fewer parameters, but they also have MREs that are worse than those of our scaling laws. Upon your remark, we have added autoscale to the experiment shown in Fig. 4 of the paper, where we examine the test MRE as a function of the number of training runs. We run the experiment for the LLM with 6 domains. For any number of training runs, Autoscale performs worse than our scaling laws. Its lower number of coefficients is not enough to counterbalance the fact that it does not model the loss as a function of h as accurately as our laws.
>
> We will include the full figure in the paper, here are a few datapoints where we report the test MRE percents:
>
> |Method| 4 training domain weights | 8 training domain weights| 16 training domain weights | 32 training domain weights|
> |---------|---------|---------|---------|---------|
> | Additive law (ours)| **1.90**|**0.53**|**0.27**|**0.21**|
> | Autoscale|2.50|1.25|0.95|0.88|
>
> This means that, even in cases where one has limited compute and can only conduct a few training runs (e.g., 4), our scaling law provides a better explanation of the loss behavior than Autoscale: there is no "*computational overhead of getting the necessary data for fitting more parameters*".
>
> We will add this important figure to the paper.
>
> | To substantiate the claim regarding the learning rate, it would be beneficial to include an evaluation using a different scheduler.
>
> Our evaluations already encompass two widely used schedulers in the literature: constant and cosine decay. Both of them  are used to train state-of-the-art LLMs and to fit scaling laws. We believe that adding more schedulers would only incrementally improve the paper.
>
> We hope that our answer clarifies our contributions.

---

### Official Review · Reviewer_4cfG · 2025-07-03

**Clarity:** 2
**Significance:** 3
**Originality:** 3
**Rating:** 3
**Confidence:** 3

**Summary:**

This paper investigates pre-training hyperparameters for Neural Multimodal Models (NMM) and Large Language Models (LLM) to explore scaling laws. It details the specific configurations and optimization strategies used, including model sizes, learning rates, optimizers, batch sizes, and data augmentations. A key aspect of the work is the comparison of their proposed scaling laws, which account for model scale (N), data (D), and data mixture (h), against existing laws that primarily consider data mixture alone. The authors demonstrate that their additive scaling law consistently achieves lower Mean Relative Error (MRE) on both training and testing datasets compared to alternative models. Furthermore, the paper explores the concept of optimal domain weights, revealing that training on a pure target domain often leads to optimal weights concentrated in that domain's corner of the probability simplex. The analysis of Jensen-Shannon distance suggests that, in general, training on a mixture different from the target mixture can be more effective for minimizing loss.

**Questions:**

1. How to treat the data changes? Does this method have to retrain multiple models if new data/domain is added or some domains are deleted?
2. Does this method allow for different model architectures? For example, using multiple small models to fit the scaling law while using another larger model with a different architecture to train. Would it still get a benefit?

**Ethical Concerns:**

["NO or VERY MINOR ethics concerns only"]

**Final Justification:**

I'm not sure if it's okay to ignore the modality difference in such problem. So I decrease my confidence.

**Limitations:**

See weakness.

**Quality:**

2

**Strengths And Weaknesses:**

**Strength**

1. The paper integrates Scaling Laws into Data Mixing. The paper introduces a principled method to determine optimal data mixtures. These laws predict model loss based on model size ($N$), training tokens ($D$), and data mixture ($h$), a comprehensive approach that surpasses prior work considering only mixture for fixed $N$, $D$. Fitting these laws on small-scale runs allows accurate extrapolation of performance and optimal mixture identification for larger, unseen models.
2. Consideration of Multi-modality in Data Mixing. This paper uniquely extends data mixture optimization to multimodal models. It applies scaling laws to Neural Multimodal Models (NMMs) that process interleaved text and image tokens.

**Weakness**
1. The difference between single-modality and multi-modality in the data mixing problem is not very clear. I don't get how to treat these two cases differently in the formulation. Or they share the same method while just can apply to two different settings in applications. Could you please elaborate on this more?
2. One domain can only include one type of data in multi-modality setting. In my understanding, one domain in the paper can only include one type of data like text or paired. How to process it if one domain includes multiple types of data like both text and paired?
3. Missing the analysis of computational cost. While the authors claim that _"Only 10-20 runs are needed for determining the scaling laws."_ in line 229. It's unclear what sizes of models it requires and how much computational cost it uses.
4. It would be better if compare with other relative data mixing baselines like Data Mixing Laws [1] and RegMix [2]. These two papers also use multiple models with different sizes to predict domain weights for larger models. Could you please clarify the difference compared with them, including the prediction accuracy and computational cost?
5. Missing downstream task evaluation. The paper only uses loss as a metric while the performance on downstream tasks is closer to reality.

[1] Ye, Jiasheng, et al. "Data mixing laws: Optimizing data mixtures by predicting language modeling performance." _arXiv preprint arXiv:2403.16952_ (2024).

[2] Liu, Qian, et al. "Regmix: Data mixture as regression for language model pre-training." _arXiv preprint arXiv:2407.01492_ (2024).

---

> ### Author Rebuttal · Authors · 2025-07-29
>
> We thank the reviewer for these thoughtful and constructive comments, which will improve the final version of the paper. Below we address each point in detail:
>
> ### Clarification regarding “The authors demonstrate that their additive scaling law consistently achieves lower Mean Relative Error (MRE) on both training and testing datasets compared to alternative models”
>
> We would like to clarify that we are the first to propose a joint (N, D, h) law, and the alternative methods are other baselines that we propose. We have a discussion in the “Related Work” regarding the limitations of other works.
>
> ### Single-Modality vs. Multi-Modality Settings
>
> Our framework, described in Section 2.1, defines training as sampling from a mixture of data sources (called domains), with mixture weights denoted by h. This formulation is agnostic to the type of data in each domain. In our LLM experiments, all domains are text, while for multi-modal experiments (e.g., NMM training), and similar to how multimodal models are are commonly trained, each domain happens to correspond to a different modality (e.g., text,  paired image-text, interleaved image-text). To strengthen this more, we also validate our scaling laws for large vision encoder training, where domains come from different image-caption datasets.
>
> For that experiment, we use AIMv2 style models / scaling (Fini 2024), with models of size 90M to 531M for fitting the laws, and up to 1.1B for evaluation. The four domains are paired image-caption datasets: (1) noisy alt-text sourced from the Internet (COYO-700M and DFN2B), which provide large-scale real-world image-text pairs with varying levels of noise and quality; (2) HQ-ITP-1, a high-quality dataset containing 134 million samples; (3) HQ-ITP-2, another high-quality dataset comprising 400 million samples; and (4) synthetic data, consisting of  recaptioned versions of DFN2B and HQ-ITP-2.
> We obtain the following low MRE % for the four domains:
>
> |Domain | Eval MRE % (Additive / joint)|
> | -------- | ------- |
> |1 | 1.20 / 0.63|
> |2|  6.19 / 5.94|
> |3|  2.02 / 1.18|
> |4|  0.79 / 1.06|
>
> Which gives another different setup where our scaling laws work, with another definition of a "domain".
>
> Overall, our method does not require separate handling of single- vs. multi-modal settings; both are unified under the same scaling law formulation.
>
> ### Domains with Mixed Modalities
>
> In the paper we consider each domain to have the same modalities, either only text, or only paired image-text or interleaved.
>
> Upon your remark, we have also run an experiment in the NMM setting where a domain is a mixture of text and image captions. In that experiment, the third domain consists of data sampled randomly from either the text or the image captions domains, with equal probability for each. The first and second domains are like in the main experiments, they are respectively image-caption and interleaved data. We then fit the scaling laws using these new three domains, where the goal is to predict the loss on the image caption, interleaved, and text domains. The evaluations MREs we obtain are as follows:
>
>
> |Domain | Eval MRE % (Additive / joint)|
> | -------- | ------- |
> |Image caption: | 0.88 / 0.96|
> |Interleaved|   0.53 / 0.40|
> |Text | 0.43 / 0.34|
>
>
> Which are very similar to the results in table 2 with the original domains. We will add this important discussion to the text; this further strengthens our argument that these scaling laws are universal, regardless of the definition one uses for a “domain”.
>
>
> ### Computational Cost Analysis:
>
> This is an interesting point. We use the following assumptions: Flops are computed as 6ND, and we use a number of training runs that is enough to get to a low MRE, as per Fig.4. We report the training costs for the model sizes shown in table 1.
> NMM (using 6 runs per sizes): small scale runs cost: 6.9e21 flops. Cost of one 8B run: 7.8e21 flops (Note that we could have trained this model for longer, linearly increasing the computational costs. Also, reducing the number of training tokens to 60B for the small scale runs does not significantly impact the eval MRE, while it reduces the small scale flops to 4.1e21).
> LLM (using 10 runs per size): small scale runs cost: 2.9e20 flops. Cost of one 3B run: 1.1e21 flops, which is approximately five times higher.
> We will add these important numbers in the final version of the paper.
>
>
> ### Comparison with Related Work ([1] and [2]):
>
> We already compare against Data Mixing Laws [1] extensively in Appendix B. Their approach models performance for fixed (N,D), while ours introduces a joint law over (N,D,h), enabling extrapolation across mixture compositions and scales. In comparable settings where we fix N and D, our method achieves lower MRE than theirs.
>
> RegMix [2], similarly, proposes a linear law over h for fixed (N,D), without modeling scaling behavior. We compared against RegMix in the same setup as in appendix B.
> We get the following test MREs for the 4 domains:
>
>
> | Scaling law formula | Wikipedia | GitHub|Stackexchange|Pg-19|
> |------|------|------|------|------|
> | Additive, fixed (N, D) | **0.18**| **0.19**|**0.18**|**0.12**|
> | Regmix | 1.31|1.35|0.92|0.89|
>
> The test MREs for regmix are very far from the ones we get with the proposed scaling law. We will add these results to the paper.
>
>
> We already compare our laws to that of Bimix in Table 10 in Appendix B. The Bimix scaling law does not predict how the loss evolves with N while our law does, and even in the fixed N setting, our laws have a far lower test MRE.
>
>
> ### Downstream Task Evaluation:
>
> We agree that downstream performance is ultimately what matters. To address this, we added a new study with larger-scale LLMs (up to 7B parameters), where we show that the model trained with our predicted optimal mixture, achieves the best results.
> The study is as follows:
> We train llms on 7 domains from slimpajama, and fit the scaling laws with small scale runs (<1B parameters). We then train large 7b models with 160B tokens with the optimal domain weights to get the best loss on average for the 7 domains (model A) , or to get the best loss on OpenHermes, a high quality domain (model B). We compare these models to models trained on the base distribution (Model C) for slimpajama and the uniform distribution (Model D) across the 7 domains. To get the best performing models, we train them with a cosine schedule, while the small scale runs were done with constant lrs. In terms of loss, Model A achieves the best loss on average over the domains, and model B achieves the best loss on openhermes. This shows that the optimal weights found by our laws still yield superior performances at large scales. We then score those four 7B models on standard benchmarks. Model B, targeting a high quality domain, yields the best scores, followed by model A, while D and C are comparable. This demonstrates that the optimal weights found at smaller scale to optimize the next token prediction loss also yield good performances on downstream tasks. The downstream tasks' results are as follows (CORE is the median of low variance benchmarks (Li 2024)):
>
> |Model | CORE median score| MMLU|
> | -------- | ------- |--------|
> |Optimal weights for avg| 56% | 32% |
> |Optimal weights for OpenHermes|  **58%** |**37%**|
> |Base | 52% | 25% |
> |Uniform | 53% | 30% |
>
> That said, we acknowledge that the precise relationship between pretraining loss and downstream task success is still an open question in the scaling law literature, see Schaeffer (2024); Mizrahi (2025); Gadre (2025); Bhagia (2024) for recent attempts.
>
> *Li, Jeffrey, et al. "Datacomp-lm: In search of the next generation of training sets for language models." NeurIPS (2024): 14200-14282.*
>
> *Schaeffer, R., Schoelkopf, H., Miranda, B., Mukobi, G., Madan, V., Ibrahim, A., Bradley, H., Biderman, S. and Koyejo, S., 2024. Why Has Predicting Downstream Capabilities of Frontier AI Models with Scale Remained Elusive?. arXiv preprint arXiv:2406.04391.*
>
> *Mizrahi, D., Larsen, A.B.L., Allardice, J., Petryk, S., Gorokhov, Y., Li, J., Fang, A., Gardner, J., Gunter, T. and Dehghan, A., 2025. Language Models Improve When Pretraining Data Matches Target Tasks. arXiv preprint arXiv:2507.12466.*
>
> *Gadre, S.Y., Smyrnis, G., Shankar, V., Gururangan, S., Wortsman, M., Shao, R., Mercat, J., Fang, A., Li, J., Keh, S. and Xin, R., Language models scale reliably with over-training and on downstream tasks. ICLR 2025.*
>
> *Bhagia, A., Liu, J., Wettig, A., Heineman, D., Tafjord, O., Jha, A.H., Soldaini, L., Smith, N.A., Groeneveld, D., Koh, P.W. and Dodge, J., 2024. Establishing task scaling laws via compute-efficient model ladders. arXiv preprint arXiv:2412.04403.*
>
> ### Q1.
>
> We follow standard pretraining practice and assume a fixed set of domains. That said, our method is flexible to removing domains--this is handled by setting the corresponding weight in h to zero. To add a new domain, additional training runs involving that domain would be needed, but all previous runs remain valid datapoints (with the new domain weight set to zero).
>
> ### Q2.
>
> This is an interesting question. To our knowledge, even in standard scaling law literature (e.g., Chinchilla, Llama), scaling is typically studied within a fixed architecture family, varying only model width/depth. We follow the same approach here.
> Empirically, changes in architecture should only shift the scaling law coefficients (i.e., constants A,B,E), but leave the overall functional form unchanged. Formalizing this is an interesting direction for future work. This phenomenon has been documented in the past (Mayilvahanan et al, 2025): the data mixture has a stronger impact than the model architecture on scaling.
>
> *Mayilvahanan, P., Wiedemer, T., Mallick, S., Bethge, M. and Brendel, W., LLMs on the Line: Data Determines Loss-to-Loss Scaling Laws. In Forty-second International Conference on Machine Learning (2025).*
>
>
> We thank you again for your review, and we hope that we have clarified any grey areas.

---

> > ### Comment · Reviewer_4cfG · 2025-08-04
> >
> > Thanks for the authors' detailed responses and additional experiments!
> >
> > However, I'm not fully convinced by being agnostic to different modalities in the domain.
> > - [3, 54] in the paper indicate that inter-modality interactions are a crucial scaling factor that the paper's unified law may not fully capture.
> > - For the vision encoder training experiment, it's actually a simpler case than Table 2 because all domains have the same modality setting (all of them include image-text paired data).

---

> ### Author Response · Authors · 2025-08-05
>
> Dear reviewer,
>
> We are unsure what you mean by "being agnostic to different modalities in the domain".
>
> ### Clarification on the meaning of domains and the validity of our law
>
> In the new version of the paper, we demonstrate that our scaling laws formulas work for these different types of domains:
> - Text domains for LLMs (domains from the Pile or SlimPajama)
> - Text-image domains for LVMs (domains from different sources/data quality)
> - Multimodal domains where each domain corresponds to one modality for NMM (text, captions, interleaved)
> - Multimodal domains where one domain corresponds to a mixture of two modalities (text + image caption; caption; interleaved)
>
> In all these cases, our scaling laws extrapolate well, achieving a low MRE on unseen large-scale models. This means that they accurately fit the data and predict the behavior of unseen models on unseen data mixtures, for all the above definitions of "domains".
> The fact that we can use our scaling laws to then predict the optimal domain weights, and that training a model with those weights outperforms all other models, is another reassuring news regarding the validity of our law, for all the above definitions of "domains".
>
> We also would like to stress that we only claim that our scaling laws are valid for the above domains: we tested text, image captions and interleaved data. Exploring these scaling laws for other modalities such as audio or video would be a very interesting future research direction.
>
>
> ### Link with existing literature
>
>
>
> Indeed [3, 54] fit scaling laws for fixed data mixtures and report different coefficients for the scaling laws depending on the mixture. This *does not* contradict our findings. There are two crucial differences with our work, which we will explain with more clarity in the final text:
> - These works fit the scaling law on the training loss, which is itself a function of the mixture weights $h$. In our paper, we rather estimate the loss on each individual domain $L^1,\dots, L^k$ where $k$ is the number of domains. The training loss is then obtained as a byproduct, by computing the weighted sum: $L^{\mathrm{train}} = h_1L^1 +\dots + h_k L^k$. [3, 54] directly fit a scaling law to estimate  $L^{\mathrm{train}}$ for a fixed mixture $h$.
> - Indeed, [3, 54] report different scaling law coefficients for different mixtures. This is in line with our scaling laws formulas: one can interpret eq. 4 and 5 as one scaling law for each mixture $h$, which has different coefficients as $h$ changes, exactly like in [3, 54]. Hence, we believe that our scaling laws capture the "inter-modality interactions" that you mention; the proof being "in the data": they extrapolate very well to larger scale models.
>
> ### Different powers
>
> In [54], the authors report different powers $\alpha, \beta$ as the mixture is changed, while both our scaling laws have fixed powers $\alpha, \beta$. There is no contradiction here, because our scaling laws model the loss of each domain, while [54] models the training loss, that is, the weighted sum of the domain losses.
>
> For instance, our additive law predicts that, for a fixed $h$ and model size $N$, the losses on the three domains behave as a function of $D$, up to a constant, as:
> -  $L^{text}(D) = \frac{22000}{D^{0.48}} + C$
> -  $L^{caption}(D) = \frac{10000}{D^{0.44}} + C$
> -  $L^{interleaved}(D) = \frac{55000}{D^{0.52}} + C$
>
> Therefore, as in [54], when one wants to estimate the *training* loss for the mixture $0.1, 0.1, 0.8$, our scaling laws predict that it behaves as  $L^{train} =  0.1 \times \frac{22000}{D^{0.48}} +  0.1 \times \frac{10000}{D^{0.44}} + 0.8\times \frac{55000}{D^{0.52}} + C$, which is not a power law with a single power, yet is well approximated[*] by the power law  $L^{train} \simeq  \frac{40000}{D^{0.51}} + C$ for $D$ in the range $10B-100B$.
>
> Now, if one considers a different mixture $0.4, 0.5, 0.1$, one gets a new approximation  $L^{train} \simeq  \frac{16000}{D^{0.45}} + C$ for $D$ in the range $10B-100B$.
>
> This leads to different powers in the training loss, like in [54], even though the per domain powers are constant.
>
> Thanks to your remark, we will add this important discussion to the text.
>
>
> We hope that our answer has clarified our points; please let us know if anything is still unclear.
>
> [*] This approximation is obtained by fiting a scaling law of the form $\frac{A}{D^{\alpha}}$ to approximate the training loss.

---

### Official Review · Reviewer_G7h4 · 2025-07-04

**Clarity:** 3
**Significance:** 4
**Originality:** 3
**Rating:** 6
**Confidence:** 3

**Summary:**

This paper offers a new scaling law study for data mixtures (simplex vector h) as a function of model size N and data size D. They experiment with two different types of scaling curves, additive and multiplicative (depending on the dependency between params/data and datamix weights). They first show an efficient way to fit these parameters with limited training runs, and then show that the estimation extrapolates sufficiently well to large-scale training. They provide ablations using pre-training MLLMs and LLMs with different scale of domain mixtures.

**Questions:**

- The paper does not investigate the case where there are lot of domain mixtures - like on the order of 100. It is very common in post-training to have diverse set of mixtures (unlike pre-training with homogenous mixtures). They authors can add their thoughts on this.

- I am curious why did the authors choose inverse scaling law for h in eq 4? For N,D - it is intuitive because as we scale models and data, the loss should go down. But what might explain the fact that the coefficients are inversely related to the loss (in Eq 4 and 5)?

- Minor: Looks like D and N have interchanged coeffiencts between Eq 3 and 4?

- Minor: Having grid lines in Fig 3 might make it more clear to understand the slope of observed vs. predicted loss curves.

**Ethical Concerns:**

["NO or VERY MINOR ethics concerns only"]

**Final Justification:**

I am fairly convinced that this paper addresses a very non-trivial problem (scaling laws for pre-training datamixes) with effective observations (validation if the proposed law + importance of text<->image data). I'd stay with my rating.

**Limitations:**

The paper adequately discusses several limitations in sec 8, several of which I concur with.

**Paper Formatting Concerns:**

None.

**Quality:**

3

**Strengths And Weaknesses:**

Strengths
--------------

1. The paper considers a very important problem of robustly estimating the datamix weights for large-scale pre-training using small-scale runs with impressive results.
2. The actual modifications made to the standard scaling law is minimal (easy to adopt and optimize) yet effective (fits most training regimes well) validating the strengths of the hypothesis.
3. Extensive empirical study (using ~300k GPU hours) on multimodal as well as text LLMs with several important conclusions and observations which go beyond current scaling study (like importance of text data for Large LLMs in fig 5).
4. The paper provides insights into more efficient optimization strategies as the #domains scale up, which might be important for post-training stages which involve several different mixtures and domains.
5. The paper honestly states several limitations, which is important to drive future work in this area.

Weaknesses
-----------------

Please refer to the questions below.

---

> ### Author Rebuttal · Authors · 2025-07-29
>
> Dear reviewer,
>
> We thank the reviewer for the very positive feedback, thoughtful questions and helpful suggestions.
>
>
> ### Scalability to 100s of domains
>
> As you mention, our current method does not scale efficiently to such large numbers of domains, since, as a rule of thumb, the number of experimental runs required scales approximately linearly with the number of mixture components. However, it is  significantly more efficient than existing alternatives based on exhaustive trial-and-error, which becomes prohibitively expensive at scale. We agree it’s important to be upfront about this limitation, and we will clarify it in the final version. On the practical side, in order to find a middle ground and still use our laws, one can group similar domains to reduce the size of the search spaces (e.g. Coding, Math, General Text …). Identifying a more efficient algorithm that can scale to 100s of microdomains is a great question for future research.
>
>
> ### Choice of Inverse Scaling in Eq. 4
>
> Our choice was motivated by a natural extension of classical scaling laws, which model loss as a power-law function of data size and model size. While it’s intuitive for loss to decrease with increasing N or D, the situation with h is more nuanced. We would like to clarify that in our formulation, the exponents are free to take on negative values, so the coefficients themselves are not necessarily inversely proportional to the loss. We selected this formulation because it yielded the best empirical performance among several plausible extensions, and we will further clarify this rationale in the revised text. For an empirical comparison with other formulations please refer to Sec 6./Tab. 3 and Appendix B. We also observed that the relative ratios had more impact than the absolute ones (e.g, 10% VS 5% has more effects than 30% VS 25%), which is well captured by a value of |\gamma|<1.
>
>
> ### Minor: Coefficient Swap Between Eq. 3 and 4
>
> Thank you. We will fix this.
>
>
> ### Minor: Grid Lines in Figure 3
>
> We agree, adding grid lines will improve clarity in interpreting the slopes. We'll update the figure accordingly.
>
>
> We thank you again for your review, and we hope that we have clarified any grey areas.

---

> > ### Comment · Reviewer_G7h4 · 2025-08-05
> > **Comment**
> >
> > I am fairly convinced that this paper addresses a very non-trivial problem (scaling laws for pre-training datamixes) with effective observations (validation if the proposed law + importance of text<->image data). I'd stay with my rating.

---

### Note · Authors · 2025-08-13

Dear AC and reviewers,

Thank you for your thorough comments and engagement throughout this review process.

The reviews from **reviewers QVnK and G7h4** are strongly positive, citing praise for the work as "very relevant", tackling "a very important problem" with "impressive results", that our law is "easy to adopt and optimize yet effective", and that we conduct an "extensive empirical study" / " The experimental setup is quite thorough".

**Reviewer 4cfG** raises the following weaknesses, all of which have been addressed in our rebuttal:
- 1, 2) a lack of clarity regarding modalities.
This has been clarified in our rebuttal.

- 3) a missing analysis of computational costs.
This has been added.

- 4) an allegedly missing comparison with Data Mixing Laws, a missing comparison with RegMix.
A comparison to Data Mixing Laws was already present in the Appendix. Regmix has been added, and is outperformed by our proposed method.

- 5) missing downstream evaluations
We have included downstream evaluations, and added a new experimental setup with LLM sizes up to 7B parameters, showing superior eval performance by our proposed method.

After one round of back and forth, this reviewer stopped replying and did not submit the acknowledgement of our rebuttal.

**Reviewer mQkL** raises the following weaknesses, all of which have been addressed in our rebuttal:

- 1) a lack of comparison with other works, (Autoscale, Doremi, Doge) that estimate optimal domain weights
We have added a comparison to Autoscale, which is outperformed by our proposed method.
We do not compare to Doremi and Doge because they require heavy modification of the training loop and are costly to run. We have also clarified that while optimal domain weights estimation is a key contribution of our work, it is not the only contribution; for instance, Autoscale, Doremi and Doge do not model the loss as a function of N, D, h.

- 2) a lack of downstream evaluations:
These have been added.

- 3) a lack of theoretical justifications.
We agree. The goal of our work, like most scaling studies, is to find and use a reasonable law that extrapolates well. Establishing a theoretical justification is important future work.


We hope that these remarks provide a clear summary of the discussion phase, and we thank you again for taking the time to assess our work.

---

### Decision · Program_Chairs · 2025-09-17

**Decision:**

Accept (poster)

**Comment:**

This paper propose a systematic method to determine the optimal data mixture for any target domain using scaling laws. The reviewer ratings are mixed. Reviewers acknowledge the strengths of the paper, such as it is tackling a very important problem of robustly estimating the datamix weights, the method is easy to adopt and optimize yet effective, the empirical study is extensive and thorough, and multi-modality is considered in data mixing. At the same time, concerns are also raised, such as missing comparison with related work (e.g., Data Mixing Laws, RegMix, Autoscale, Doremi, Doge), a lack of clarity regarding modalities, missing downstream evaluations. The authors have provided comprehensive response to address the concerns. Although the reviewers have not confirmed whether the concerns have been fully addressed or not and still maintain negative ratings, I thought they are answered to some extent. Considering this, we recommend an acceptance to this paper.